# Increased nitrous oxide emissions from global lakes and reservoirs since the pre-industrial era

Ya Li [1,2,3], Hanqin Tian [4] ✉, Yuanzhi Yao[5], Hao Shi [1], Zihao Bian [2,6], Yu Shi [2,7], Siyuan Wang[1,3], Taylor Maavara [8], Ronny Lauerwald [9] & Shufen Pan[2,4,10]

Lentic systems (lakes and reservoirs) are emission hotpots of nitrous oxide ($N_2O$), a potent greenhouse gas; however, this has not been well quantified yet. Here we examine how multiple environmental forcings have affected $N_2O$ emissions from global lentic systems since the pre-industrial period. Our results show that global lentic systems emitted $64.6 \pm 12.1$ Gg $N_2O$-N yr$^{-1}$ in the 2010s, increased by 126% since the 1850s. The significance of small lentic systems on mitigating $N_2O$ emissions is highlighted due to their substantial emission rates and response to terrestrial environmental changes. Incorporated with riverine emissions, this study indicates that $N_2O$ emissions from global inland waters in the 2010s was $319.6 \pm 58.2$ Gg N yr$^{-1}$. This suggests a global emission factor of 0.051% for inland water $N_2O$ emissions relative to agricultural nitrogen applications and provides the country-level emission factors (ranging from 0 to 0.341%) for improving the methodology for national greenhouse gas emission inventories.

Nitrous oxide ($N_2O$) is a potent greenhouse gas, with ~273 times the warming potential of carbon dioxide on a 100-year time horizon, and also contributes to stratospheric ozone destruction[1–3]. Nitrogen (N) processes in inland waters, as a critical component of the global N cycle, are gaining recognition for their important contribution to $N_2O$ emissions through nitrification and denitrification[4,5]. These emissions, expressed in carbon dioxide ($CO_2$) equivalents, will offset ~4% of the land carbon sink[6]. Several preceding studies have been dedicated to assessing the magnitude of $N_2O$ emissions from inland waters on regional and global scales[5,7,8]. However, the global estimates are still weakly constrained, particularly for lentic systems such as lakes and reservoirs.

Sizeable human activities have contributed to a notable increase in anthropogenic N loads that are transported from land to lentic systems, thereby playing a significant role in $N_2O$ emissions originating from these systems[8–10]. However, based only on sparse and unevenly distributed local measurements, most previous estimates on $N_2O$ emissions from lentic systems are varied by approximately four-fold (160.00-583.00 Gg N yr$^{-1}$)[5,11,12]. Furthermore, human-induced $N_2O$ emission from lentic systems are implicitly incorporated, as the indirect agricultural $N_2O$ emissions, into the recent national $N_2O$ emission inventory from the United Nations Framework Convention on Climate Change (UNFCCC), which is calculated based on anthropogenic N additions and global mean emission factors[13,14]. Nevertheless, the use of constant and linear emission factors in emission inventory fails to capture the spatial variability of $N_2O$ emissions from lentic systems[8] and cannot dynamically attribute them

[1]State Key Laboratory of Urban and Regional Ecology, Research Center for Eco-Environmental Sciences, Chinese Academy of Sciences, Beijing 100085, China. [2]International Center for Climate and Global Change Research, Auburn University, Auburn, AL 36849, USA. [3]University of Chinese Academy of Sciences, Beijing 100049, China. [4]Center for Earth System Science and Global Sustainability, Schiller Institute for Integrated Science and Society, Department of Earth and Environmental Sciences, Boston College, Chestnut Hill, MA 02467, USA. [5]School of Geographic Sciences, East China Normal University, Shanghai 610000, China. [6]School of Geography, Nanjing Normal University, Nanjing 210023, China. [7]College of Urban and Environmental Sciences, Peking University, Beijing 100871, China. [8]School of Geography, University of Leeds, Leeds LS2 9JT, UK. [9]Université Paris-Saclay, INRAE, AgroParisTech, UMR ECOSYS, Palaiseau 91120, France. [10]Department of Engineering, Boston College, Chestnut Hill, MA 02467, USA. ✉e-mail: hanqin.tian@bc.edu

to environmental changes, such as climate warming and agricultural N application. This limitation hinders the accurate estimation of $N_2O$ emissions from lentic systems at the regional level, consequently impacting the precision of national $N_2O$ emission inventories[13]. Considering the anticipated rise in terrestrial N loads to inland waters[15], there is a pressing need for a more mechanistic research framework to enhance our understanding of N cycling within aquatic environments and to refine the estimation of $N_2O$ emissions from lentic systems.

Considering the strong correlation between terrestrial N loads and $N_2O$ emissions in aquatic systems, previous modeling studies have been dedicated to predicting $N_2O$ emissions from lentic systems based on spatially explicit terrestrial N input[8,16]. However, a short-coming of existing models is that $N_2O$ production is represented as a function of dissolved inorganic nitrogen (DIN) availability, and when nitrate content reaches zero, denitrification will ceases produce $N_2O$. Nevertheless, in nature, denitrifying bacteria continue to denitrify $N_2O$ in absence of nitrate, and this leads to a decrease of $N_2O$ levels in the water, and ultimately the system can function as $N_2O$ sinks. This occurs in low-DIN systems, particularly in tropical lakes[17] and tropical rivers with a relatively low human impact[18,19]. Additionally, these studies represent single-point snapshots in time and have not fully integrated the temporally-evolving dynamically coupled N cycles of terrestrial-aquatic continuum from a mechanistic perspective, limiting their ability in representing the response of $N_2O$ budgets of lentic systems when the watershed environment experiences significant changes (such as climate change and land management). Some studies have revealed that global changes, including climate change, land use change, and atmospheric N deposition, have a substantial influence on N cycling in lentic systems[20-22]. Hence, it is imperative to develop a dynamic mechanistic model that incorporates intricate environmental changes and integrates multiple N processes for regional and global assessments.

Benefiting from our past modeling efforts in simulating the dynamic riverine $N_2O$ emissions[23], we incorporated the $N_2O$ sub-model of lentic systems with significant improvement in water transporting and the associated biogeochemical processes. This integration forms a comprehensive stream-river-lake-reservoir corridor within the framework of the Dynamic Land Ecosystem Model (DLEM) to represent the dynamic interaction of three N species (DIN, dissolved organic N and particulate organic N) across the terrestrial-aquatic continuum. We compared the simulated inland water $N_2O$ fluxes and aquatic nitrate concentration with the observations around the globe to showcase the good performance of our model, with $R^2$ values exceeding 0.6 and Nash-Sutcliffe efficiency coefficient (NSE) exceeding 0.5. Then, we assess global $N_2O$ emissions from lakes and reservoirs ($N_2O$-LR) from the pre-industrial period to the recent decade (1850-2019) and examine their sensitivity to environmental changes. Derived from two global lentic system datasets (HydroLAKES and GRanD database)[24,25], we categorize lakes and reservoirs into "small" or "large", depending on their upstream catchment area and the connectivity to subnetwork flows or main river channels in this study. Those with an upstream catchment area greater than the area of a 0.5° grid cell are defined as large lentic systems, and linked to the main channel corridor; correspondingly, the remaining lentic systems with a smaller upstream area are defined as small lentic systems, and linked to the subnetwork corridor. Furthermore, for management purposes, we quantified emission factors for global countries, which is applicable to national greenhouse gas emission inventories for estimating agricultural contributions to $N_2O$ emissions from inland waters. Here, we define the agricultural $N_2O$ emission factor for inland waters ($EF_{Ag}$) as the proportion of agricultural $N_2O$ emissions from inland waters (attributed to synthetic fertilizer and manure application) relative to the total agricultural N additions.

## Results

### The spatiotemporal patterns of global lake and reservoir $N_2O$ emissions during the 1850s-2010s

Our simulation utilized high-resolution data for global lentic systems that is derived from the HydroLAKES and GRanD datasets[24,25]. The total surface area covered by these lentic systems is 2,900,538 $km^2$, where lakes account for 85% (50% for large lakes and 35% for small lakes) and reservoirs accounting for 15% (14% for large reservoirs and 1% for small reservoirs), respectively (Supplementary Table 1). Driven by these lentic system data, our study shows that the estimated $N_2O$-LR in the 2010s (2010-2019) totaled $64.6 \pm 12.1$ Gg N $yr^{-1}$ (mean ± standard deviation of the annual average), with 88% from lakes ($56.9 \pm 10.6$ Gg N $yr^{-1}$) and 12% from reservoirs ($7.8 \pm 1.5$ Gg N $yr^{-1}$) (Fig. 1a). The decadal mean $N_2O$-LR increased significantly ($p < 0.01$) from the pre-industrial period (the 1850s, $28.6 \pm 6.8$ Gg N $yr^{-1}$) to the 2010s, with an average increase rate of 0.2 Gg N $yr^{-1}$ (Fig. 1a). The most notable increase in total $N_2O$-LR was found from the 1940s to the 1980s, with an annual increase rate of 0.4 Gg N $yr^{-1}$. Since the 1980s, the increase rate in $N_2O$-LR had slowed down to a rate of 0.3 Gg N $yr^{-1}$ during the 1980s-2010s (Fig. 1a).

$N_2O$ emissions intensities in small lentic systems were higher than those from large lentic systems. Specifically, in the 1850s, the $N_2O$ emission per unit area from small lakes was $13.2 \pm 0.1$ mg N $m^{-2}$ $yr^{-1}$, whereas it was $10.4 \pm 2.9$ mg N $m^{-2}$ $yr^{-1}$ for large lakes. Due to the larger area of global large lakes, $N_2O$ emissions from these lakes contributed larger to the overall $N_2O$ emissions than small lakes in the 1850s, with $N_2O$ emission shares of large and small lakes being 53% and 47%, respectively (Fig. 1b). In the 2010s, $N_2O$ emission per unit area from small reservoirs ($128.2 \pm 22.3$ mg N $m^{-2}$ $yr^{-1}$) is three times higher than that from small lakes ($30.6 \pm 5.0$ mg N $m^{-2}$ $yr^{-1}$), which is followed by large lakes ($17.6 \pm 3.8$ mg N $m^{-2}$ $yr^{-1}$) and large reservoirs ($10.0 \pm 2.2$ mg N $m^{-2}$ $yr^{-1}$). Therefore, despite the small lentic systems comprising only 36% of the total surface area, they contributed to 55% of the total $N_2O$ emissions in the 2010s. There had been a 133% increase in $N_2O$ emissions from small lakes during the 1850s-2010s, which is approximately twice that of large lakes (Fig. 1b). From the 1850s to the 2010s, the total increase in $N_2O$-LR was primarily attributed to small lakes (50%), followed by the large lakes (29%), the large reservoirs (11%), and the small reservoirs (10%) (Fig. 1c).

In the pre-industrial period (the 1850s), North America (8.3 Gg N $yr^{-1}$), East Asia (5.5 Gg N $yr^{-1}$), and South America (4.5 Gg N $yr^{-1}$) were hotspots for $N_2O$-LR (Fig. 2a), collectively accounting for 64% of the global $N_2O$-LR (Supplementary Fig. 1). From the 1850s to the 2010s, around 75% of the increasing global $N_2O$-LR was from northern mid- to high-latitudes (30°N-60°N) (Fig. 2c). During the 2010s, $N_2O$-LR showed the peaks in northern mid- to high-latitudes (30 °N-60°N) and the tropics (5°N-5°S) (Fig. 2c). The regions with intensive agricultural activities including East Asia (16.3 Gg N $yr^{-1}$), North America (14.0 Gg N $yr^{-1}$), Europe (8.6 Gg N $yr^{-1}$) and Africa (7.1 Gg N $yr^{-1}$), contributed 71% of total $N_2O$-LR (Supplementary Fig. 1), becoming hotspots of $N_2O$-LR in the 2010s (Fig. 2a). At the regional level, the amount of $N_2O$-LR from Europe, East Asia, South Asia, Southeast Asia, and West/Central Asia have increased more than twofold (Supplementary Fig. 1), with up to eightfold increases in some grids since the 1850s (Fig. 2b). $N_2O$-LR in most regions increased significantly especially after the 1960s (Supplementary Fig. 2). The increasing rate of $N_2O$ emissions from small lentic systems in most regions, except Africa, South Asia, and West/Central Asia, is 2–8 times higher than the rate in large lentic systems (Supplementary Fig. 2). The increase in rates of $N_2O$ emission from small lentic systems in Europe (0.03 Gg N $yr^{-1}$), North America (0.02 Gg N $yr^{-1}$), and East Asia (0.04 Gg N $yr^{-1}$) were much higher than those in other regions.

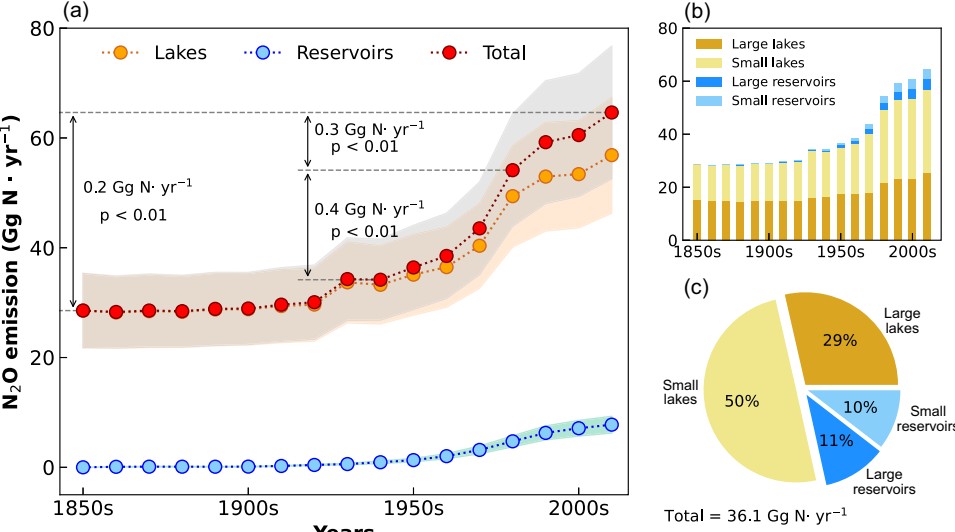

**Fig. 1 | Changes in N$_2$O emissions from global lakes and reservoirs since the pre-industrial period. a** Total N$_2$O emissions from global lakes and reservoirs (red dotted line) and N$_2$O emissions from global lakes (orange dotted line) and global reservoirs (blue dotted line) during the 1850s-2010s; the orange, blue, and gray bands represent the uncertainty (mean ± standard deviation of the annual average) of N$_2$O emissions. **b** N$_2$O emissions from different lentic systems. **c** The relative contribution of different lentic systems to overall increase in global N$_2$O emissions from lakes and reservoirs during 1850s-2010s.

## Relative roles of natural and anthropogenic environmental factors on lentic system N$_2$O emissions over time and regions

Globally, N$_2$O-LR were enhanced by climate change, land use change, and agricultural N additions, but reduced by elevated atmospheric CO$_2$ concentration (Fig. 3). It should be noted that the initiation of reservoir simulations on the grid cells is determined by the year of reservoir construction. As a result, when conducting factorial experiments, the effect of reservoir construction will undoubtedly be included in the influence of environmental changes on N$_2$O-LR. Since N$_2$O emissions from reservoirs constituted 12% of N$_2$O-LR in the 2010s, we assume that the impact of reservoir construction represents only a minor portion of the overall impact. From the 1850s to the 1940s, climate change and land use change collectively contributed to 85% of the increase in N$_2$O-LR, which was primarily attributed to higher terrestrial N input driven by global warming and increased agricultural activities. During the 1940s-1980s, the global N$_2$O-LR experienced a threefold increase, when agricultural N addition, including synthetic fertilizer and manure application, contributed 68% of this increase, while the contribution of climate change and land use change was only 22%. In the 1980s to the 2010s, agricultural N addition remained the dominant driver for increased N$_2$O-LR, with its contribution on global increased N$_2$O-LR being twice higher than that of climate change and six times higher than that of land use change. Nevertheless, the increased magnitude of N$_2$O-LR originating from agricultural N addition, climate change, and land use change were comparatively lower than that in the previous period. In the 2010s, agricultural N addition is the primary factor responsible for enhancing N$_2$O-LR in most regions, except in Africa and Russia where climate change remains dominant. Agricultural N addition contributes up to 60% of increased N$_2$O-LR in Southeastern Asia, Southern Asia, Europe, Eastern Asia, and West/Central Asia (Supplementary Fig. 3). Notably, from the pre-industrial period to the recent decade, the elevated atmospheric CO$_2$ concentration accelerated plant growth and N uptake, and thus inhibited terrestrial N loss to lentic systems and N$_2$O-LR. The inhibitory effect of elevated atmospheric CO$_2$ concentration on N$_2$O-LR has been on the rise from 0.9 Gg N yr$^{-1}$ in the 1850s-1940s to 3.3 Gg N yr$^{-1}$ in the 1980s-2010s, which resulting in a 52% reduction in N$_2$O-LR and nearly offset the promoting effect induced by climate change and land use change.

We also found that small lentic systems showed greater response to global changes (Supplementary Fig. 4). Induced by climate change and human disturbance, N$_2$O emissions from small lentic systems increased by 211% (3.5 Gg N yr$^{-1}$) during the 1850s-1940s and by 247% (12.6 Gg N yr$^{-1}$) during the 1940s-1980s. In comparison, the rates of increase of N$_2$O emissions from large lentic systems were slower, with 66% (1.9 Gg N yr$^{-1}$) increase during the 1850s-1940s and 185% (8.8 Gg N yr$^{-1}$) increase during the 1940s-1980s. Although the area of small lentic systems only accounts for 36% of the total lentic system area, they show a greater response to environmental change. Specifically, during the 1850s-1940s, the strong responses of N$_2$O emissions from small lentic systems were attributable to climate change (63%), agricultural N addition (90%), atmospheric N deposition (72%), and increased atmospheric CO$_2$ concentration (-54%), and outweighed the responses of large lentic systems, constituting more than half of the total responses. During the 1940s-1980s, while the influence of climate change on N$_2$O emissions from small lentic systems diminished (decreasing from 63% to 36%), the responses of small lentic systems to agricultural N additions (68%), atmospheric N deposition (54%), and increased atmospheric CO$_2$ concentration (-64%) still amounted to over half of the total responses. In the period of the 1980s-2010s, N$_2$O emissions from small lentic systems have greater responses to natural disturbances such as climate change (56%) and increased atmospheric CO$_2$ concentration (-63%) (Supplementary Fig. 4).

## Estimate of global inland water N$_2$O emissions: integrating updated riverine estimation

The transport and transformation of N in inland waters have significant cascading effects. For instance, the nutrient N is transported to lakes and reservoirs through upstream rivers; as a potent reactor of N species, lakes or reservoirs have a significant impact on nutrient N exported further to the downstream rivers. Therefore, in this study, we present an updated estimation of N$_2$O emissions from global streams and rivers, encompassing the processes of lentic systems that were excluded from our previous estimates (Supplementary Fig. 5). In the pre-industrial period (the 1850s), global streams and rivers emitted 83.8 ± 22.8 Gg N yr$^{-1}$ of N$_2$O into atmosphere. Since the pre-industrial era, there has been a significant growth in global riverine N$_2$O emissions, particularly from the 1960s to the 2010s, exhibiting a linear growth rate of 26.2 Gg N per decade (Supplementary Fig. 5). The remarkable increase of global riverine N$_2$O emissions can be attributed

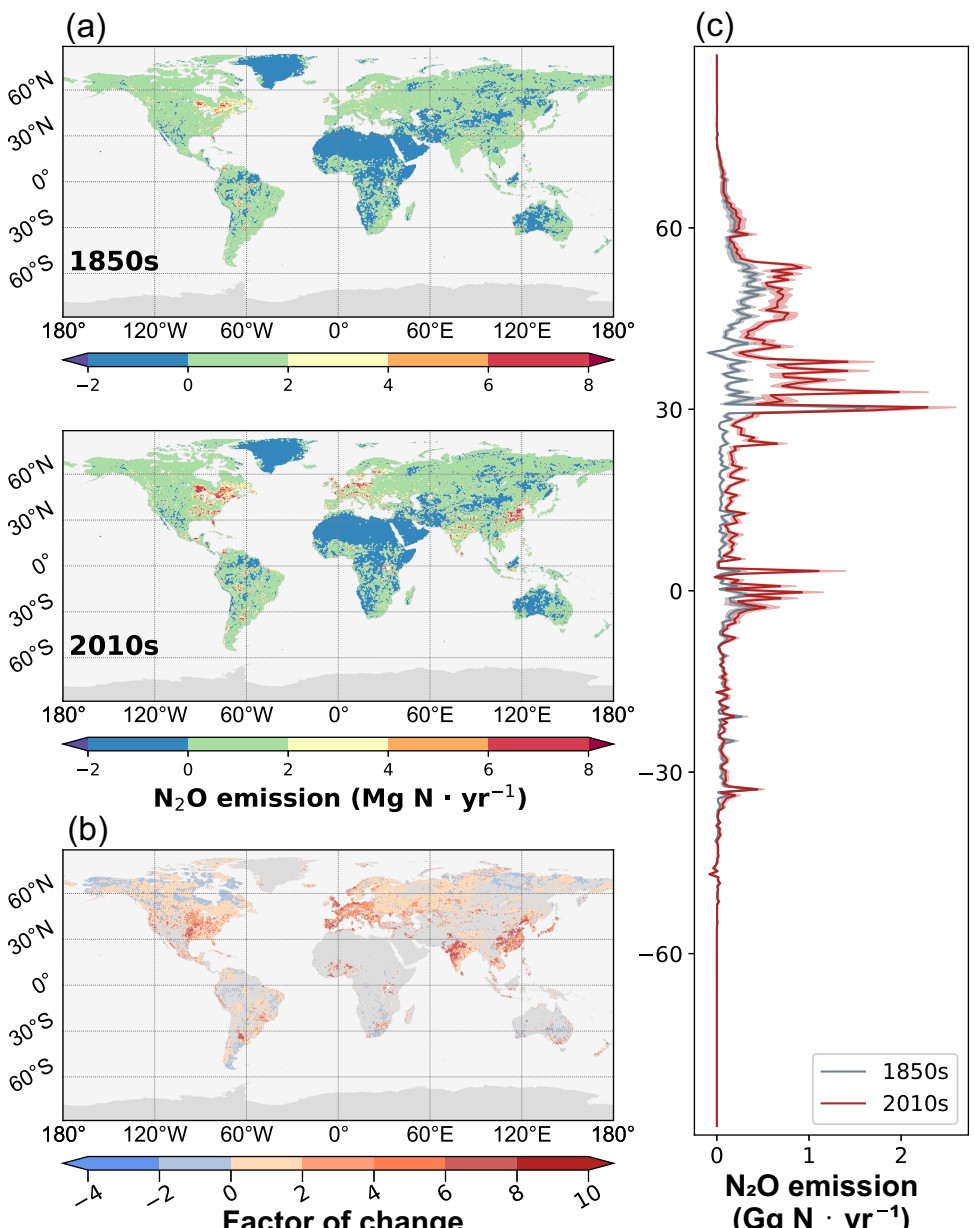

**Fig. 2 | Spatial pattern of N₂O emissions from global lakes and reservoirs. a** The spatial pattern of N₂O emissions from lakes and reservoirs in the 1850s and the 2010s, respectively; **b** the changed rates of the N₂O emissions from global lakes and reservoirs in the 2010s relative to 1850s; **c** latitudinal distribution of N₂O emissions from lakes and reservoirs in the 1850s (blue line) and the 2010s (red line); the red and gray bands represent the uncertainty (mean ± standard deviation of the annual average) of N₂O emissions. Figure made using the Matplotlib Basemap Toolkit library[79] in the Python programming language (version 3.10.9, from Anaconda version 2023.3).

primarily to the application of agricultural N fertilizer and N manure (Supplementary Fig. 6). In the recent decade, global riverine N₂O emissions reached 254.9 ± 46.2 Gg N yr⁻¹, marking a twofold increase from the estimated levels in the 1850s (Supplementary Fig. 5).

Combining the updated amount of riverine N₂O emissions, the estimated N₂O emission from global inland waters in the 2010s was 319.6 ± 58.2 Gg N yr⁻¹ (Fig. 4). In the 2010s, riverine N₂O emissions constitute 80% of the total emissions from inland waters worldwide, with N₂O-LR being accountable for the remaining 20%. N₂O emissions from global inland waters have substantially increased by 207.2 Gg N yr⁻¹ since the 1850s, of which the increased N₂O-LR induced by anthropogenic perturbation contributes to 17% of the total increase of N₂O emissions from inland waters.

## Dynamic anthropogenic emission factors of N₂O emissions from inland waters

Regarding the finding that agricultural activities are responsible for the accelerated growth in N₂O emissions from inland waters, here we propose the revised $EF_{Ag}$ to improve the overall clarity of anthropogenic indirect N₂O emissions within national greenhouse gas emission inventories. Our factorial experiments reveal the dynamic contribution of agricultural N additions on N₂O emissions from inland waters (Supplementary Table 2 and Table 10). From the 1850s to the 1910s, manure application resulted in negligible N₂O emissions from inland waters (Fig. 5a). In the early years of synthetic N fertilizers being applied in agricultural practices (around the 1920s), agricultural N additions led to a slight increase in N₂O emissions from inland waters,

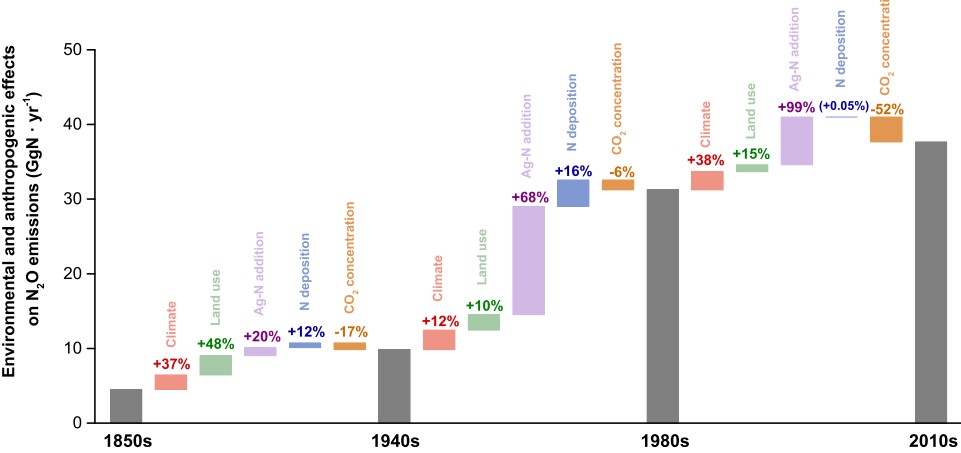

**Fig. 3 | The relative contributions of environmental and anthropogenic factors to N₂O emission changes from global lakes and reservoirs over different time periods.** The gray bars show mean decadal N₂O emissions from global lakes and reservoirs induced by five forcing factors. The colored bars and their percentages represent the relative contribution of each forcing factor to the net change of total effect for the corresponding periods. Ag-N addition represents the agricultural nitrogen additions, which includes synthetic fertilizer and manure application.

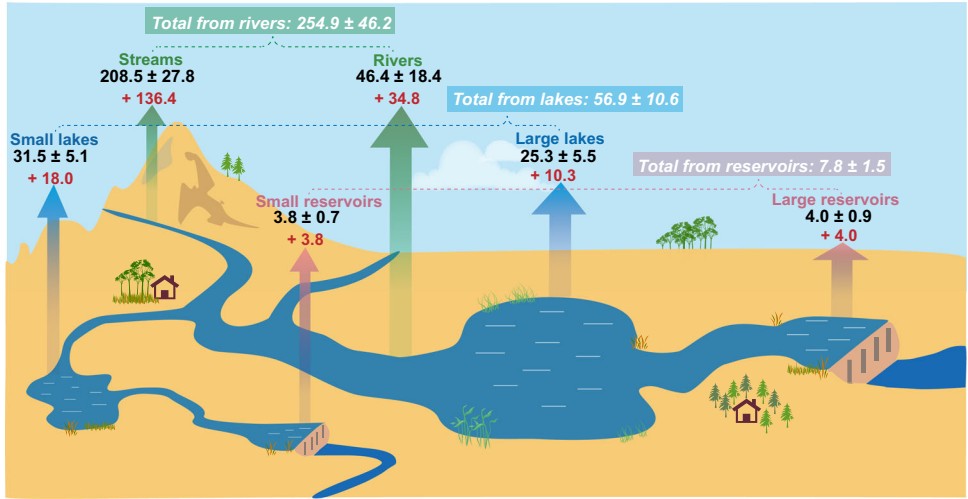

**Fig. 4 | Global inland water N₂O emissions for the 2010s.** The colored arrows represent N₂O emissions as follows: green, emissions from streams and rivers; blue, emissions from small and large lakes; pink, emissions from small and large reservoirs. The colored numbers represent N₂O fluxes as follows: bold black numbers, the emissions in the 2010s; bold red numbers, the increased emissions during the 1850s-2010s; italic white numbers with green, blue, and pink background colors represent total N₂O emissions from rivers, lakes, and reservoirs, respectively. The unit for all numbers is Gg N yr⁻¹. The graph was drawn using the Adobe Illustrator 2020.

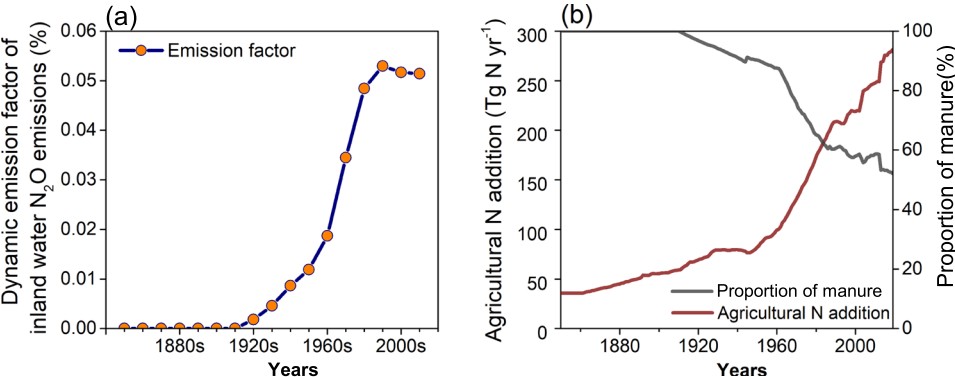

**Fig. 5 | Global mean emission factors of N₂O emissions from inland waters. a** Dynamics of mean global emission factors for inland water N₂O emissions. **b** Dynamic amount of agricultural nitrogen addition and the proportion of manure in agricultural nitrogen addition.

with a $EF_{Ag}$ of only 0.002%. By the 1990s, the fraction of synthetic N fertilizer in agricultural N addition increased from around zero to 45% (Fig. 5b). More importantly, global mean $EF_{Ag}$ constantly raised and reached 0.053% in the 1990s (Fig. 5a; Supplementary Table 2). However, despite the ongoing growth in agricultural N addition, $EF_{Ag}$ in recent decades are lower than that in the 1990s. In the 2010s, the global mean $EF_{Ag}$ stand at 0.051%, with national-level $EF_{Ag}$ ranging from 0.000 to 0.341%. $EF_{Ag}$ of thirty-nine countries are exceeding global mean $EF_{Ag}$ in the 2010s (Supplementary Table 3).

## Discussion

In the past, observation-based studies have provided a rough referred range for N$_2$O-LR (160.0-380.0 Gg N yr$^{-1}$, 30.0-70.0 Gg N yr$^{-1}$, and 400.9-583.0 Gg N yr$^{-1}$ for lakes, reservoirs, and total lentic systems, which were equivalent to 68.6-163.0 Tg yr$^{-1}$, 12.9-30.0 Tg yr$^{-1}$, and 172.0-250.1 Tg yr$^{-1}$ of CO$_2$ emissions, respectively, using a GWP of 273 over 100 years; see Supplementary Table 4)[5,9,11,12]. However, most of observation-based studies were linearly upscaled relied on a small number observation and usually had poor constraint on estimates due to the uneven distribution of observational data in time and space. Specifically, high N$_2$O fluxes at individual sites may result in over-estimation of the entire region[26]. In recent years, the modeling studies that consider the mechanistic processes of N$_2$O emissions have been developed, provided the new methodology for estimating N$_2$O-LR[8,16]. In this study, we present an amount of N$_2$O-LR that is relatively low than the observation-based estimates and close to a previous model estimate[16]. Our advantage, however, lies in considering the relation-ship between watershed environmental changes and aquatic N$_2$O emissions, allowing us to simulate the dynamic changes in N$_2$O-LR driven by climate change and human activities. Our updated estimate of riverine N$_2$O emissions exhibits a reduction of 37.4 Gg N yr$^{-1}$ in comparison to the previous one[23], equivalent to 14% of the newest estimate in this study. The disparity can be attributed to the newly incorporated module for lentic systems, which intercepted a portion of N during transport. This interception results in a reduction in the amount of N received by downstream streams or rivers connected to these lentic systems and corresponding N$_2$O emissions. Simultaneously, modeling of in-river transformation and losses of N will also restrict N transfer to lentic systems. This emphasizes the significance of complete nutrient transport processes along the terrestrial–aquatic continuum in modeling studies to constrain global N$_2$O emission from inland waters. Furthermore, we estimate N$_2$O emissions from global inland waters to be 319. 6 ± 58. 4 Gg N yr$^{-1}$, which falls within the range of previous estimation (204.9-1270.0 Gg N yr$^{-1}$)[5,7,8,11,27,28]. Our estimated global inland water N$_2$O emission is only half of that estimated by Beaulieu, et al.[7] which may be attribute to the discrepancy on the uptake velocities and terrestrial N input to inland waters. For instance, we assumed the lower denitrification uptake velocity ranging from 3E-08 to 2E-06 m s$^{-1}$ and simulated lower DIN input of 48 Tg N yr$^{-1}$ (TN input of 89 Tg N yr$^{-1}$) in the 2010s, compared to their reported deni-trification uptake velocity of 8E-08 to 1E-05 m s$^{-1}$ and DIN input of 90 Tg N yr$^{-1}$, respectively. However, their study likely has led to an overestimation[23], as their estimation was based on higher emission factors and DIN loads compared to those indicated by other studies[29–34]. Furthermore, Beaulieu, et al.[7] assumed that all N$_2$O pro-duced in water was emitted, a point that has been argued due to its potential to overestimate aquatic N$_2$O emissions especially when considering the effect of water residence time[8]. A latest synthesis[28] homogenized global scale estimates to present N$_2$O emission of 204.9 (157.8-375.5) Gg N yr$^{-1}$ from global inland waters, which is close to our estimate. Another recent modeling study by Wang et al.[27] reporting higher inland water N$_2$O emissions of 0.4 Tg N yr$^{-1}$ in 1900 and 1.3 Tg N yr$^{-1}$ in 2010. However, in their study, the oversight of the seasonal emission fluctuations under the yearly modeling time step and the potential inhibited effect of elevated atmospheric CO$_2$

levels on terrestrial N availability and subsequent N loss may introduce significant uncertainty into their estimates. Moreover, for some N sources included in their study such as aquaculture and wastewater, the existing datasets still fall short in providing us with accurate quantification on the effect of these sources on global inland water N$_2$O emissions. Therefore, future development of modeling input data will help reduce uncertainties in the model estimates.

This study demonstrates that increase in N$_2$O-LR since the pre-industrial period are primarily caused by anthropogenic N loads, with modulation from climate change, land use conversions, and elevated atmospheric CO$_2$ concentrations. Inland waters receive a large amount of N from agricultural practices, additionally atmospheric N deposition affects >90% of the surface area of lakes worldwide[35,36]. Direct inputs of external N stimulate N$_2$O-LR, as shown by studies in oligotrophic to eutrophic lentic systems[37–39]. A previous study[40] reported that the increase in terrestrial N loads can be attributed to enhanced N mineralization due to increasing temperature when compared to N uptake, thus explaining the increase in N$_2$O-LR due to climate change in our study. Additionally, several local observational studies indicate that temperature increases are likely to stimulate the nitrifying and denitrifying microbial activity in water systems and increased N$_2$O emissions[41,42]. It is worth noting that the increase in N$_2$O-LR caused by changes in terrestrial N loads may outweigh the effect of temperature on the control of microbial metabolism[41,43]. Land use conversions play a significant role in altering soil N cycling and promoting terrestrial N losses[44,45], thereby contributing to increased N$_2$O-LR. This phenom-enon appears to be particularly pronounced in Africa. With rapid population growth and increased demand for agricultural land and wood products, large areas of natural forests in sub-Saharan Africa have been deforested or converted to agricultural land[46]. The con-version of natural forests which serve the function of protecting and bonding soils, to other artificial land-use types caused severe nutrients loss from soil[47], leading to a significant increase in N$_2$O-LR. In addition, the increasing frequency of extreme precipitation events observed in Africa[48] has accelerated the soil erosion by water on these human-disturbed regions. In contrast to other factors, the elevated atmo-spheric CO$_2$ concentration exhibited negative effects on N$_2$O-LR. Ele-vated atmospheric CO$_2$ levels have long been considered as an important driver in reducing soil N availability[49]. The fertilizing effect of atmospheric CO$_2$ enhances carbon assimilation by plants and increases the foliar carbon-to-nitrogen ratio[50]. The higher carbon-to-nitrogen ratios in leaf litter could promote microbial N uptake and reduce net N mineralization in soil[49,51,52], consequently limiting N$_2$O-LR due to the reduced terrestrial N loads. It would be worth pointing out that while N$_2$O-LR is predicted to be leveling off, no leveling off of increases in global soil N$_2$O emissions have been noted, instead they appear to be accelerating[1].

Our findings reveal significant differences in N$_2$O emissions and their sensitivity to environmental changes between small and large lentic systems. We found that small lentic systems play a crucial role as hotspots for N$_2$O emissions within the global lentic system. This can be attributed not only to the higher effectiveness of small lentic systems in N removal processes[53], but also to their advantageous geographic position, enabling them to intercept a sizable portion of terrestrial N loads prior to reaching downstream large lentic systems. Conse-quently, this interception prevents the captured N from contributing to the nitrification and denitrification occurring in the downstream large lentic systems. The shallow African lakes, with considerable organic matter deposition on the sediment, sustain high benthic denitrification rates, as suggested by Borges, et al.[17]. Since there is a general relation between lake surface area and depth[54], that standpoint also supports the higher N$_2$O emission rates from the small lakes in our study. However, if the inorganic nitrogen concentration cannot sup-port high denitrification rates on sediment, the N$_2$O produced in the water column will be absorbed by sediment to fuel benthic

denitrification[17]. This explains the contrasting findings in Borges, et al.[17] which showed the $N_2O$ undersaturation in the shallow African lakes. Reservoirs, distinguished from lakes by long water residence times and their location in densely populated areas with substantial human-induced N loads, are widely recognized as aquatic $N_2O$ emission hotspots[9,16,55]. Although previous work has highlighted the contribution of longer water residence times and an anoxic bottom water column on promoting denitrification within large reservoirs[56], it is crucial to recognize the significant role played by upstream small reservoirs in limiting denitrification in downstream large reservoirs. Their effective retention of inorganic N substantially reduced N concentrations in the outflow (reductions can even exceed 50%)[57–60], thereby restricting the nitrification and denitrification within downstream large reservoirs. Compared to large lentic systems, small lentic systems are characterized by higher importance of terrestrial N inputs relative to surface area and volume. Therefore, changes in terrestrial N inputs caused by global change showed a stronger impact on $N_2O$ emissions from small lentic systems. Our findings underscore the significance of small lentic systems in the N cycle of inland water systems and their potential role in mitigating global $N_2O$ emissions in response to future anthropogenic activities.

Agricultural activities play a crucial role in influencing $N_2O$ emissions from inland waters[1,23]. The existing UNFCCC national greenhouse gas emission inventories therefore employed the recommended methodology by Intergovernmental Panel on Climate Change (IPCC) to estimate anthropogenic inland water $N_2O$ emissions resulting from managed soil leaching. However, these estimations are usually characterized by large spatial uncertainty since observed data used for determining the emission factors in the IPCC's report are limited, and inadequately reported in non-Annex I countries[13,14]. Based on our simulation, we recommend a global averaged $EF_{Ag}$ value of 0.051% as the proportion of agricultural N addition emitted as $N_2O$ through inland waters in the current environmental condition. Rather than a constant value, the ratio would change with the varied agricultural management or environmental conditions. Until the early 20th century, agricultural N additions solely consisted of manure enriched in organic N and carbon[61] (Fig. 5b). However, following the invention of synthetic ammonia technology, the increased use of synthetic fertilizers in agriculture led to a greater fraction of inorganic N in total terrestrial N loads, which subsequently enhanced $EF_{Ag}$ for $N_2O$ emissions from inland waters. Over the past two decades, despite the increasing agricultural N additions, EFs have decreased due to the suppressive effect of elevated atmospheric $CO_2$ concentrations on soil N loss. After discounting emission factors collected from previous studies based on 24% of the proportion leached from agricultural N additions[14], we find that our results yield lower estimates than those reported in most of previous studies[7,11,14,30] and align with the lower boundary of the range estimated by Maavara, et al.[8] (Supplementary Table 5). This discrepancy can be attributed to the representation of coupled terrestrial and aquatic processes in the model utilized in this study, which allowed for the isolation of inland water $N_2O$ emissions by agricultural N additions specifically. In contrast, previous estimates, which used aquatic nitrate concentration without separating environmental impacts, would likely include the effects of other environmental factors in their EFs. Although the estimate by Beaulieu, et al.[7] have separated the impact of agricultural N additions, their estimate is based on the observation of headwater streams (generally thought to have higher emission rates), potentially leading to an overestimated EF for global inland waters. Furthermore, our results indicated that EFs reported in previous studies may not be suitable for assessing inland water $N_2O$ emissions under future climate and human activity scenarios. Hence, we advocate for future research to adopt mechanistic models to accurately evaluate $N_2O$ emissions from inland waters. Meanwhile, the national-level $EF_{Ag}$ presented in this study can still provide governments and local managers with intuitive and easy-to-use parameters for estimating inland water $N_2O$ emissions in current scenarios.

Improving the representation of biogeochemical processes in mechanistic models and enhancing the quality of measured and driving data can help reduce uncertainties in simulating $N_2O$ emissions from lentic systems. Rigorous mutual verification between the process-based model and field observations are crucial for reducing the estimated gap. To better constrain the $N_2O$-LR estimates in our simulations, we compared simulated terrestrial N loading with measured data across natural and agricultural land worldwide, as terrestrial N loading is the primary substrate for $N_2O$ production in inland waters. Nevertheless, point source N inputs from industrial wastewater were not included in the current simulation, thus our estimates may underestimate $N_2O$ emissions in watersheds receiving substantial nutrient release. In addition, another source of uncertainty of our study is the representation of global lentic systems. The HydroLAKES and GRanD databases used here do not include lentic systems with surface areas <0.1 $km^2$, thus we likely underestimate $N_2O$ emissions from small ponds, which are considered as an important $N_2O$ source[11]. Although we included additional $N_2O$ emissions from newly constructed reservoirs over time, we did not consider the impact of lake area changes on $N_2O$ emissions due to the limited availability of dynamic lake surface data. Considering that a recent study demonstrated the increasing trend of global lake area in recent decades[62], it is likely that our study gives a conservative estimate for $N_2O$-LR. In future research, improving data quality or using multiple input datasets will help address the remaining uncertainties for global models.

## Methods

### Dynamic Land Ecosystem Model-Terrestrial-Aquatic Continuum (DLEM-TAC)

To quantify $N_2O$ emissions from global inland waters (rivers, lakes, and reservoirs), we use a process-based coupled terrestrial-aquatic model, which is built on the framework of the Dynamic Land Ecosystem Model (DLEM). DLEM-TAC is a fully distributed, process-based land surface model which couples the major land processes (terrestrial hydrology, plant phenology and physiology, soil biogeochemistry) and aquatic dynamics (lateral transport and in-stream biogeochemistry)[23,63–65]. The land component of DLEM-TAC explicitly simulates the carbon, N, and water fluxes between plants, soil, and atmosphere, and the surface and drainage runoff and N load from land module are used as the input of the riverine module. The simulated N load includes DIN, dissolved organic nitrogen (DON), particulate organic nitrogen (PON), and runoff, which serve as the major inputs to the aquatic module.

The DLEM-TAC aquatic module calculates lateral water transport and the associated aquatic biogeochemical processes by adopting a scale-adaptive scheme (Supplementary Fig. 7). The water transport scheme couples hillslope flow, subnetwork flow, and main channel flow with a grid cell as subgrid processes, which allows the representation of small scale physical and biogeochemical processes at larger spatial scales. The subnetwork flow, which is conventionally known as the 1st-5th order rivers in the 0.5° grid cell[66,67], receives water from hillslope flow and drains into the main channel. Based on our previous study[23], we coupled the lentic systems into the subnetwork and river routing to form a river-lake-reservoir corridor in this so improved model. Furthermore, lentic systems where the upstream area is smaller than the area of the 0.5° grid cell are classified as small lentic systems, and assigned to the linked subnetwork corridor; correspondingly, the remainders with the upstream catchment area larger than the grid area are classified as large lentic systems, and assigned to the linked main channel corridor. The surface area of large/small lakes or large/small reservoirs are shown in Supplementary Table 1. The incoming water and nutrient flows of sub grid lakes and reservoirs linked to subnetworks depend on their upstream area obtained from the high-resolution dataset[24], which determines the fraction of flows

from hillslope and subsurface that are intercepted. The water of a linked river-lake-reservoir corridor of a subnetwork drains to lakes and reservoirs first, and the outflow rate of lakes and reservoirs is determined based on the predefined residence time obtained from the global lake dataset[24,68,69]. The aquatic N module was developed based on the scale adaptive water transport scheme[23,70], including lateral transport, decomposition of organic matter, particulate organic matter deposition, nitrification, and denitrification.

Following our previous work referring to the representation of water transport and biogeochemical cycling, we developed an inland water $N_2O$ module within the aquatic biogeochemical component of the DLEM-TAC framework[71] (Supplementary Fig. 8). The net fluxes of dissolved $N_2O$ (including physical and biogeochemical processes) in inland waters are estimated as:

$$(\Delta M_{N2O})/\Delta t = F_a + Y_{water} + D - R - E \qquad (1)$$

where $M_{N2O}$ is the total mass of dissolved $N_2O$ in inland waters (g N), $\Delta t$ is the time step, $F_a$ is advective $N_2O$ fluxes (g N d$^{-1}$) (Supplementary Text 1), $Y_{water}$ is the $N_2O$ production within inland waters (g N d$^{-1}$) (Supplementary Text 2), $D$ is the dissolved $N_2O$ from rainfall to inland waters (i.e. deposition) (g N d$^{-1}$) with an initial concentration equal to the atmospheric equilibrium $N_2O$ concentration, $R$ is the flux from $N_2O$ reduction (g N d$^{-1}$) to dinitrogen gas (Supplementary Text 3), and $E$ is $N_2O$ efflux (g N d$^{-1}$) through the air-water interface (Supplementary Text 4).

### Input data

The driving data of the DLEM-TAC include the climate variables, atmospheric $CO_2$ concentration, land use change, N deposition, N fertilizer, and manure application with a spatial resolution of 0.5° × 0.5°. The daily climate variables (precipitation, mean temperature, maximum temperature, minimum temperature, and shortwave radiation) were obtained from the CRUNCEP dataset (https://vesg.ipsl.upmc.fr) for 1901-2019. Climate data during 1850-1900 cycled early 20th century (1901-1920) climate[72]. Annual atmospheric $CO_2$ concentration from 1900-2019 was obtained from the NOAA GLOBALVIEW-$CO_2$ dataset (https://www.esrl.noaa.gov). The annual land use change data were derived from a potential natural vegetation map (synergetic land cover product) and a prescribed cropland area dataset from the history database of the global environment v.3.2 (HYDE 3.2, ftp://ftp.pbl.nl/hyde). The data of N fertilizer, manure N application, and N deposition were obtained from Tian, et al.[73].

In the aquatic module, the required channel dataset included channel slope, channel width, and channel length generated from the HydroSHEDs dataset[70,74] and DDM30 dataset[75]. The flow direction and distance data were obtained from the Dominant River Tracing (DRT) dataset[76]. For modeling water dynamics in lakes and reservoirs, we generated 0.5° grid level surface water area, upstream area, volume, depth, and average residence time for lakes based on the HydroLAKES dataset[24], while the GRanD v1.01 database provided the same information for reservoirs[69].

### Simulation protocol

The DLEM-TAC simulation includes three steps: equilibrium run, spin-up run and two transit runs, one with dam operation closed, and another one with dam operation open. First, the equilibrium run is required to obtain the initial and steady state condition of carbon, N, and water pools at the pre-industrial level in each grid cell[77]. In this step, we held all the driving forces such as climate data, atmospheric $CO_2$ concentration, land use data, and N additions consistent with the first year's data we used in the simulation. Second, we conducted a 30-year spin-up run by randomly selecting climate data within the 1850s[78]. This step can alleviate the disturbance of driving data changes in the transit run. Then we conducted the natural flow simulation with the

dam model temporarily deactivated (no dams), and all the driving forces change over time. After the natural flow simulation, we set up a management flow simulation with the dam module open, because the dam module needs natural flow in the previous run as model input[68]. Based on natural flow, the management flow simulation for newly constructed reservoirs over time were conducted starting from the constructed years provided by GRanD v1.01 database. To evaluate the model performance, we compared the simulated $N_2O$ emission to 106 observed $N_2O$ fluxes from global inland waters including lakes, reservoirs and rivers. In addition, we also validated the simulated aquatic nitrate concentration. The simulated results agreed well with the observation with the value of $R^2$ above 0.6 and NSE above 0.6 in most cases (Supplementary Fig. 9).

To quantify the effects of environmental factors such as climate change, atmospheric $CO_2$ concentration, land use change, N deposition, and agricultural N application (fertilizer and manure) on $N_2O$-LR, we conducted other five factorial experiments though holding each environmental factor consistent with the first year of corresponding environmental data (Supplementary Table 10).

### Calculation of agricultural $N_2O$ emission factors for inland waters

In the UNFCCC national GHG emission inventories, EFs applied in estimates of agriculture-induced inland water $N_2O$ emissions are derived from the methodology provided by IPCC's report[14]. In that report, EFs produced from leaching and runoff of N addition are defined as the fraction of N leaching and runoff that is lost through $N_2O$ emissions, and further assumes that 24% of the agricultural N addition in managed land of wet climates is lost through leaching and runoff. To facilitate the calculation of agriculture-induced inland water $N_2O$ emissions in individual countries, we calculate $EF_{Ag}$ as the percentage of agriculture-induced inland water $N_2O$ emissions relative to the agricultural N additions to avoid applying a constant as the proportion of agricultural N addition loss:

$$EF_{Ag} = \frac{N2O_{inland\ water\_Ag}}{Agricultural\ N\ additions} \times 100\% \qquad (2)$$

where $N2O_{inland\ water\_Ag}$ is agriculture-induced $N_2O$ emissions from inland waters, which is calculated as the difference of inland water $N_2O$ emissions between Simulation 1 and Simulation 6 (Simulation 1 is the all-combined simulation with all the driving forces changing over time; Simulation 6 is factorial simulation experiment by holding agricultural N application at the first year, see Supplementary Table 10). The amount of N additions is obtained from Tian, et al.[73].

Raw $EF_{Ag}$ we calculated are negative between the 1850s-1910s, which can be explained by the unsaturated $N_2O$ concentrations in inland waters under the small amount of manure application. Therefore, we forced $EF_{Ag}$ to zero for the period of the 1850s-1910s. Negative $EF_{Ag}$ at specific countries were treated accordingly, which usually located in regions less affected by agriculture.

### Quantifying the uncertainty induced by terrestrial nitrogen inputs

The previous studies have identified N loads as a major source of uncertainty in inland water $N_2O$ emission estimates[16,23]. Here we evaluate the uncertainty in estimating $N_2O$-LR induced by variations in N loads. We collected 62 field datapoints of N leaching, which covers five N species ($NO_3^-$, $NH_4^+$, DON, TDN, TN) and four types of land use (cropland, forests, grassland, peatland), and validated against the simulated N leaching by DLEM-TAC (Supplementary Fig. 10). Then, we calculated the 95% uncertainty ranges of N loading using the Origin software. Finally, we determine the uncertainty range of ±22%, ±50%, ±37% and ±26% for $NO_3^-$, $NH_4^+$, DON and PON loads, respectively. We then conducted two model simulations from 1850 to 2019 with the

parameters of terrestrial loads of $NO_3^-$, $NH_4^+$, DON and PON varying ±22%, ±50%, ±37% and ±26%, respectively.

## Data availability

The data of $N_2O$ emissions from global inland waters generated in this study have been deposited in the Zenodo database (https://doi.org/10.5281/zenodo.10364781). The validated data collected from other studies are provided in the Supplementary Information file. The CRUNCEP data used in this study are freely available at https://vesg.ipsl.upmc.fr. NOAA GLOBALVIEW-$CO_2$ data used in this study are available at https://www.esrl.noaa.gov. The hydrological data required in the model are available at https://www.hydrosheds.org/products/hydrosheds (HydroSHEDs dataset), https://csdms.colorado.edu/wiki/Data:DDM30 (DDM30 dataset), https://www.hydrosheds.org/products/hydrolakes (HydroLAKES dataset), and https://sedac.ciesin.columbia.edu/data/collection/grand-v1 (GRanD v1.01 database), respectively. The map of national administrative boundaries is freely available at https://www.resdc.cn/data.aspx?DATAID=205.

## Code availability

The relevant code of this study is available from the corresponding author on request. We acknowledge the ArcMap software version 10.8 (https://www.esri.com/en-us/arcgis/products/arcgis-desktop/resources), Python version 3.10.9 from Anaconda software version 2023.3 (https://www.anaconda.com/blog/new-release-anaconda-distribution-2023-03), Adobe Illustrator 2020 (https://blog.adobe.com/en/publish/2019/11/04/adobe-illustrator-2020), and the Origin version 9.0 (OriginLab, https://www.originlab.com/index.aspx?go=Products/Origin).

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

## Acknowledgements

This study has been supported by National Natural Science Foundation of China (Grant Number: 42171463) and National Science Foundation (Grant Numbers: 1903722, 243232, and 1922687). This study contributes to the Global $N_2O$ budget Assessment sponsored by Global Carbon Project (GCP) and International Nitrogen Initiative (INI). R.L. acknowledges funding from French state aid, managed by ANR under the "Investissements d'avenir" program (ANR-16-CONV-0003). Y.Y. acknowledges funding from National Natural Science Foundation of China (Grant Number: 42371410). S.W. acknowledges funding from National Natural Science Foundation of China (Grant Number: 42171463) and from the Research Center for Eco-Environmental Sciences (RCEES), Chinese Academy of Sciences (Grant Number: RCEES-TDZ-2021-15).

## Author contributions

H.T. designed the research; Y.L. and Y.Y. performed the research and analyzed data; Y.L., H.T., Y.Y., H.S., Z.B., Y.S., S.W., T.M., R.L., and S.P. wrote the paper.

## Competing interests

The authors declare no competing interest.
