## [Peer Review File · Nature Communications]

Increased nitrous oxide emissions from global lakes and reservoirs since the pre-industrial eraReviewer #1 (Remarks to the Author):

L 55-57 : This statement is incorrect. N₂O emissions from lakes have been reported recently in using an observational approach at regional scale (Borges et al. 2022) and modelling approach at global scale (Lauerwald et al. 2019).

L77-88 : An additional short-coming of existing models is that N₂O production parameterized directly a function of DIN availability, and in the case of NO₃⁻ = 0, denitrification will no longer produce N₂O. However, in nature, denitrifying bacteria continue to denitrify N₂O in absence of NO₃⁻, and this leads to a decrease of N₂O levels in the water, and ultimately the system can function as sinks of N₂O in particular in systems with low DIN levels. This occurs in several tropical lakes (Borges et al. 2022). Actually this seems to occur quite systematically in low DIN environments, and is a generalized feature in tropical rivers with a relatively low human impact (Borges et al. 2019; Chiriboga and Borges 2023).

89-93 : It is essential to explain here what is the difference between the present work and the one of Lauerwald et al. (2019) since it seems (at least to me) to be from the same group, and a very similar model setup.

L 111 : Please explain here how you define « large » and « small » lakes/reservoirs.

L157-174: It is unclear if the effect of changes of the surface area of lakes in response of climate change are accounted in the simulations (Zhao et al. 2022; Zhou et al. 2021).

L 265 : The two cited references are for a single system. It is unreasonable to make a generalization on the response of N₂O emissions from lentic systems to warming based on two sites.

L275-296: Borges et al. (2022) suggested that N₂O in lakes could change as function of depth that modulates the coupling between surface waters and sediments. Since there is a general relation between surface area and depth (larger systems are in general deeper) (Wetzel 2001) this could also play a role in explaining differences between small and large systems.

L316-317: There have been several past papers reporting that the IPCC EF are too high. It would be useful to include these in the discussion here.

L 340 : Unfortunately the authors did not provide a Table with fluxes from observational studies. I have the impression the data from African lakes from the recent paper of Borges et al. (2022) are missing from this comparison.

L342: the r² alone is not sufficient for validation. You can have a very good correlation (high r²) and a poor validation due to a very large off-set (Y-intercept > or < 0) or over-/under-estimate (slope > or < 1) or both. So please make much more rigorous statistical evaluation of the validation.

L346: There's some room for debate. Borges et al. (2022) shows on the contrary that smaller and shallower African lakes have lower N₂O concentration than larger and deeper ones.

L 348 : do you mean seasonal or long-term changes of surface area ?

Please note that the HydroLAKES data-base in some cases over-estimates the lake surface area because some of data are quite old (HydroLAKES builds on the GLWD) and some lakes have shrunken with global warming. This is the case for arid and semi-arid climates, for instance Lake Chad, as discussed by Borges et al. (2022).

REFERENCES

Borges AV, L Deirmendjian, S Bouillon, W Okello, T Lambert, FAE Roland, VF Razanamahandry, NRG Voarintsoa, F Darchambeau, IA Kimirei, J-P Descy, GH Allen, C Morana (2022) Greenhouse gas emissions from African lakes are no longer a blind spot, *Science Advances*, 8, eabi8716, 1-17,

<https://doi.org/10.1126/sciadv.abi8716>

Chiriboga G & AV Borges (2023) Andean headwater and piedmont streams are hot spots of carbon dioxide and methane emissions in the Amazon basin, *Communications Earth & Environment*, *Commun Earth Environ* 4, 76 <https://doi.org/10.1038/s43247-023-00745-1>

Lauerwald R., P. Regnier, V. Figueiredo, A. Enrich-Prast, D. Bastviken, B. Lehner, T. Maavara, P. Raymond, Natural lakes are a minor global source of N₂O to the atmosphere. *Global Biogeochem. Cycles*, 33, 1564-1581 (2019).

Wetzel, R.G. (2001) *Limnology Lake and Reservoir Ecosystems*. Academic Press, San Diego.

Zhao G et al. (2022) *Nat Commun* 13, 3686, <https://doi.org/10.1038/s41467-022-31125-6>

Zhou W et al. (2021) *Commun Earth Environ* 2, 255, <https://doi.org/10.1038/s43247-021-00327-z>

Reviewer #2 (Remarks to the Author):

This study presents global predictions of N₂O emissions from lakes and reservoirs using a process based, coupled terrestrial-aquatic model. They find that while large lakes dominate lentic N₂O emissions, smaller lakes and small reservoirs have bigger per unit area emission. The study predicts smaller lentic N₂O emissions than most previous studies, but more than double the preindustrial emission rate.

A major strength of this study is the coupling of terrestrial and aquatic models that allow the exploration of how the major global change factors contribute to the increased N₂O emissions. The portioning of effects is very interesting. Not surprisingly, the major contributor is excess agricultural nitrogen. An interesting finding is that in recent decades, CO₂ fertilization seems to have led to reduced N₂O emissions, contributing to the slower increases in recent decades. This is based on the terrestrial model, which has its dynamics presumably described in earlier papers. It would be worth pointing out in this manuscript as part of discussion that, while aquatic N₂O emissions are predicted to be leveling off, no leveling off N₂O increases in the global atmosphere have been noted, instead they appear to be accelerating.

This manuscript emphasizes lentic N₂O emission, bringing in lotic N₂O emissions only as discussion points and as part of the emission factor calculation. This is surprising and I think a major weakness for several reasons. First, the results highlight the inland water emission factor, which includes streams and rivers, but their response to the global change factors studied in scenarios is not considered at all. The paper seems to be disconnected in this sense. Second, rivers actually dominate inland water N₂O emissions. Lentic waters are only about 10% of total inland water emissions. This is a point that I think is important to highlight. Third, river and lake processes interact as the authors discuss. However, the authors emphasize how denitrification by lakes limits N₂O emissions downstream, but not the fact that denitrification by streams and rivers limits N₂O emissions by lakes and reservoirs. This would be a compelling paper if these results were to be compared together.

Global inland water N₂O emissions are lower than most other studies, including those that incorporate the role of streams, rivers, lakes, and reservoirs, and that account for sub grid river networks (based on Beaulieu et al. 2011 which used the global aquatic N model in Wollheim et al. 2008). Little is discussed as to why the current model results in lower N₂O emissions. Is it because N loading from the landscape is less (Beaulieu model is based on annual N export from land to water in Green et al. 2004), the denitrification uptake velocity is less, the proportion of incomplete denitrification or nitrification is less, or some other reason. No parameter values are given (or calculated from the more complex model used here), but for the paper to be useful, they should be. Otherwise the results are just assertions.

I did not understand the numbers that went into the global emission factor calculation. Please

explain clearly how the 137.02 Gg N yr⁻¹ was derived.

Another weakness is that the representation of the processes that lead to N₂O emissions and key parameter values are not provided. It is mentioned (L91) that significant improvements have been incorporated, but these are not identified. In the current manuscript, only the equation for mass balance of N₂O is provided, and variable, Y_{water} , is the only mention of N₂O production. There is nothing about what controls N₂O production, which is essential for a paper like this. I had to go back to the Yao et al. 2020 paper, where more detail is given in the supplemental, but here there were errors in description of the equations (supplemental eqns 5 -9) that I was not sure what was going on. No values for uptake velocity were given here, so the reader could not evaluate why the N₂O emissions were lower than other global studies. Maybe I could go back even further to earlier papers, but I should not have to as this is essential for understanding the results of this study.

It was also not clear how the lakes at the 0.5 degree grid scale were integrated into the river network from the description. Are all small lakes within a grid cell combined into one and “placed” at the mouth of the grid cell, intercepting N that has been transported through the sub grid cell river network where denitrification and N₂O emissions can also occur? It is likely that N enters streams at the sub grid scale first, before reaching any lake at the minimum lake sizes considered. See Wollheim et al. 2008 for a very similar approach (Wollheim, W.M., C.J. Vorosmarty, A.F. Bouwman, P.A. Green, J. Harrison, E. Linder, B.J. Peterson, S. Seitzinger, and J.P.M. Syvitski. 2008b. Global N removal by freshwater aquatic systems: a spatially distributed, within-basin approach. *Global Biogeochemical Cycles*. GB2026, doi:10.1029/2007GB002963.

I appreciated the effort at validation of N₂O emissions, based on 60 lake locations, as well as the incorporation of uncertainty in the loading, which I agree is an essential issue that must be considered. But I would like to know where these observational data come from. There were no units in the figures showing these comparisons with observations. Given that this is a global model, and that nitrate concentrations likely drive N₂O emissions, in order to understand the usefulness of the model I would also like to see how well the model is tested against nitrate concentration data in streams, rivers, or lakes. Since the complete N cycle is represented, it would also be useful to compare the % of incomplete denitrification predicted to measurements, which based on LINX2 studies (Beaulieu et al 2011) demonstrated a fairly low average of 0.9% of denitrification is as N₂O. Are predictions here similar?

I also wondered about the mechanisms behind a few of the results. Why does Africa have such large increase in N₂O emissions, while ag N increase is minimal. Change is attributed to climate change in Africa, but I think it is getting drier, so why would that lead to more N₂O emissions? Table S1 reports the residence times of lakes and reservoirs. Why would small lakes have a longer residence time than large lakes (I don't think this is expected), and do reservoirs really have 10's of years of residence time?

Specific

L97. Define small and large lakes.

L106. I would suggest one fewer decimal place when reporting fluxes. I'm sure estimates are not that precise, and would make easier to read.

L162. It was not clear what this statement means.

L192. Why do smaller systems have a greater response in the model?

L195. Confusing section here.

L203. What about the role of streams and rivers upstream of lakes? Should be mentioned.

L210. This is only half of the contemporary estimate from the global model of lakes, rivers, reservoirs reported in Beualeigh et al. 2011. Can you include some discussion of why?

L247. Lake emissions should also be lower when incorporated into river network because of stream and river reductions. i.e. it works in both directions.

L277. Are small lakes hotspots not just because more effective denitrification (uptake velocity / hydraulic load), but also because they are upstream from large lakes, and therefore have greater inputs? It is not clear why residence times should be greater in small lakes, as it is likely that they will also have less watershed area. Please support this is case from literature.

L283. Longer residence times lead to more denitrification, but not more complete conversion of nitrate to N₂, at least based on this reference. This statement requires more support.

L292. Is this really an important dynamic? If so, provide some numbers.

L297. Please be clear that this includes streams and rivers, in addition to lentic waters. Since this is highlighted, stream and river results should be added to results section.

L315-317. This discounting is unclear.

L331-346. Consider putting model validation in a brief study design paragraph before the results, to increase trust in the model results.

L371. Did the sub grid cell river network account for serial transformations from 1st through 4th order? The integration of lakes is unclear – do small lakes intercept all sub grid river flow?. The global inland water N model of Wollheim et al. 2008 used a similar approach, including a statistical N removal model to account for small river dynamics.

L384. If lakes and reservoir residence times are predefined, does that mean it does not change with changes in hydrology (i.e. climate change)?

L396. More in how Ywater is controlled is really needed.

L422. Unclear what “dam operation closed” means. Is it that there are no reservoirs at all, or just input = output, or there is active management for hydroelectricity based on some rules?

L449. Does this mean that 76% of added N is harvested, stored, or denitrified on land? Isn't this part of the terrestrial model used here?

L479. The data availability is all raw input data from other sources, none of the data produced by this study. I would expect key model predictions would be shared, as well as the model validation data that was compiled.

Fig. 4. Color reference are confusing, as red represents both lake and changing emissions.

Fig 5b. Change secondary Y axis to “proportion manure”

Table S2. Sort based on Country name, or EFag.

Table S4. I'm not sure this table is worth including, given the very different assumption, with mist not relevant to the IPCC report.

Figure S2. Make legend larger to read

Table S4. CO₃ should have negative values, right?

Figure S5 and S8. Units needed, more informative caption (where are data from, how collected, ecosystem, biome, time). References to where they are from, and include in data that this study makes available.

Reviewer #3 (Remarks to the Author):

I have published quite a lot concerning GHG emissions from lakes so the literature is well-known to me. N₂O is a minor issue in lakes as it probably constitutes less than 3% of lakes' atmospheric impact so that does diminish the interest of this manuscript somewhat. I have a major concern about this manuscript that would need to be cleared up before publication anywhere, in my view. This paper is basically a lengthy modeling effort based on the premise that N limits N₂O emissions from lakes. The authors cite a couple of manuscripts as support of this idea (Maavara et al. 2018 [not 2019 as listed] and Lauerwald et al 2019) but both of these are modeling exercises and neither of them demonstrates the critical role of N in driving N₂O emissions from lakes. Instead, it is much more likely (see Delsonetro et al 2018) that N₂O emissions are driven by P (or primary production) and lake size. Most limnologists agree that N is almost always in excess in inland waters so the amount of N as substrate for N₂O emission should rarely be an issue. A credible study would be based on empirical analyses of the factors driving N₂O emissions in lakes. In its present state, this paper appears to be a thought study based on the query, "what would be happening if N limited N₂O emissions".

Some smaller suggestions are listed below

Lines 55-56. There are lots of studies of N₂O emissions in lakes and reservoirs. This statement is false or misleading.

Lines 229-230. Ranges of estimated impacts are listed as N here, not as CO₂ equivalents as in most similar studies.

Lines 232-234: "Observation-based" studies are suggested to be inaccurate compared to this model. Science normally depends on observations so the authors should make the case why their model might actually be more accurate. This seems to be somewhat non-scientific, to me.

Lines 371-372. Quite a lot is known about the global distribution of rivers and streams. For example, Downing et al. 2012. Why not use the best available estimates? In any case, I think the authors should explain why stream order (which is often criticized as having little biogeochemical relevance) would be relevant.

Line 394. The authors should better explain where variables like Fa or Y_{water} could be determined and where these came from.

Line 401. It should be clarified how one gets from modeled or simulated N transport to N₂O emissions. Especially given the paucity of global empirical data suggested in the Introduction.

Revisions of Manuscript: NCOMMS-23-28044

Title: Increased nitrous oxide emissions from global lakes and reservoirs since the pre-industrial era

Author(s): Ya Li, Hanqin Tian, Yuanzhi Yao, Hao Shi, Zihao Bian, Yu Shi, Siyuan Wang, Taylor Maavara, Ronny Lauerwald, Shufen Pan

We thank the reviewers and editors for taking the time to read and review this manuscript as well as for their constructive comments and feedback. We have carefully revised the manuscript in line with the feedback from the reviewers, and have added text and figures to the revised version of the manuscript.

We have included more details in the Method section, especially regarding model processes, underlying mechanisms, and model validations to better ground our findings related to dynamic N₂O emissions from inland waters. In addition, the more important results and information for riverine (lotic) N₂O emissions, and observed data used to validation were added in the revised manuscript and supplementary information.

The revisions are outlined below with a point-by-point response to each reviewer's comments. Please note that reviewers' comments are in *italics* while our responses are not. All text from the manuscript is colored blue, and citation numbers correspond to the reference section of this manuscript.

Point-to-Point Responses to Reviewers' Comments

Reviewer #1 (Remarks to the Author):

1. Comments: *L 55-57: This statement is incorrect. N₂O emissions from lakes have been reported recently in using an observational approach at regional scale (Borges et al. 2022) and modelling approach at global scale (Lauerwald et al. 2019).*

Response: We have corrected the statement in the revised main text:

L52-53: “However, the global estimates are still weakly constrained, particularly for lentic systems such as lakes and reservoirs.”

2. Comments: *L 77-88: An additional short-coming of existing models is that N₂O production parameterized directly a function of DIN availability, and in the case of NO₃⁻ = 0, denitrification will no longer produce N₂O. However, in nature, denitrifying bacteria continue to denitrify N₂O in absence of NO₃⁻, and this leads to a decrease of N₂O levels in the water, and ultimately the system can function as sinks of N₂O in particular in systems with low DIN levels. This occurs in several tropical lakes (Borges et al. 2022). Actually this seems to occur quite systematically in low DIN environments, and is a generalized feature in tropical rivers with a relatively low human impact (Borges et al. 2019; Chiriboga and Borges 2023).*

Response: Thank you for the great suggestion. We have added this shortcoming of the existing models in the revised main text. The specific modifications are found in L76-82.

L76-82: “However, a short-coming of existing models is that N₂O production is represented as a function of dissolved inorganic nitrogen (DIN) availability, and when nitrate content reaches zero, denitrification will ceases the production of N₂O. Nevertheless, in nature, denitrifying bacteria continue to denitrify N₂O in the absence of nitrate, and this leads to a decrease of N₂O levels in the water, and ultimately the system can function as N₂O sinks. This occurs in low-DIN systems, particularly in tropical lakes (17) and tropical rivers with a relatively low human impact (18, 19).”

Reference:

17. Borges, A. V., et al. Greenhouse gas emissions from African lakes are no longer a blind spot. *Sci. Adv.* 8, eabi8716 (2022).
18. Chiriboga, Gonzalo & Borges, Alberto V. Andean headwater and piedmont streams are hot spots of carbon dioxide and methane emissions in the Amazon basin. *Communications Earth & Environment* 4, 76 (2023).
19. Borges, Alberto V, et al. Variations in dissolved greenhouse gases (CO₂, CH₄, N₂O) in the Congo River network overwhelmingly driven by fluvial-wetland connectivity. *Biogeosciences* 16, 3801-3834 (2019).

3. Comments: *L 89-93: It is essential to explain here what is the difference between the present work and the one of Lauerwald et al. (2019) since it seems (at least to me) to be from the same group, and a very similar model setup.*

Response: Our work uses an entirely different model. In Lauerwald et al. (2019), nitrification and denitrification processes strongly depend on water residence time but ignore the critical impacts of water temperature, water velocity, substrate availability and land-water C-N interactions. In contrary, the model developed here not only have included the key impact mentioned above, but also have fully coupled the land-water C-N interactions which enables us to dynamically estimate N₂O emissions from inland waters under the influence of escalating natural and anthropological disturbances (climate changes, elevated atmospheric CO₂ concentration, land use management, atmospheric N deposition, and anthropological N application) through time.

In this paragraph, we discuss the limitation of existing models, namely their failure to effectively integrate terrestrial and aquatic systems. Building upon comment #2, we also highlight another drawback of the existing models, which lies in its incomplete depiction of the nitrogen cycle in water systems. Furthermore, in the next paragraph, we revised the description of the strengths of our model. The specific modifications are found in L76-87, L94-98.

L76-87: “However, a short-coming of existing models is that N₂O production is represented as a function of dissolved inorganic nitrogen (DIN) availability, and when nitrate content reaches zero, denitrification will ceases produce N₂O. Nevertheless, in nature, denitrifying bacteria continue to denitrify N₂O in absence of nitrate, and this leads to a decrease of N₂O levels in the water, and ultimately the system can function as N₂O sinks. This occurs in low-DIN systems, particularly in tropical lakes (17) and tropical rivers with a relatively low human impact (18, 19). Additionally, these studies represent single-point snapshots in time and have not fully integrated the temporally-evolving dynamically coupled N cycles of terrestrial-aquatic continuum from a mechanistic perspective, limiting their ability in representing the response of N₂O budgets of lentic systems when the watershed environment experiences significant changes (such as climate change and land management).”

L94-98: “This integration forms a comprehensive stream-river-lake-reservoir corridor within the framework of the Dynamic Land Ecosystem Model (DLEM) to represent the dynamic interaction of three N species (DIN, dissolved organic N and particulate organic N) across the terrestrial-aquatic continuum.”

Reference:

17. Borges, A. V., et al. Greenhouse gas emissions from African lakes are no longer a blind spot. *Sci. Adv.* 8, eabi8716 (2022).
18. Chiriboga, Gonzalo & Borges, Alberto V. Andean headwater and piedmont streams are hot spots of carbon dioxide and methane emissions in the Amazon basin. *Communications Earth & Environment* 4, 76 (2023).
19. Borges, Alberto V, et al. Variations in dissolved greenhouse gases (CO₂, CH₄, N₂O) in the Congo River network overwhelmingly driven by fluvial-wetland connectivity. *Biogeosciences* 16, 3801-3834 (2019).

4. Comments: *L 111: Please explain here how you define « large » and « small » lakes/reservoirs.*

Response: Thank you for the suggestion. In our previous study (Yao et al., 2020), we divided the simulations of river routing and the biogeochemical processes into two levels: main channel corridor at the grid cell level and subnetwork corridor within the grid cells. In our current model version, the lentic systems (lakes and reservoirs) are further connected to the river systems. Therefore, we

classified lentic systems with an upstream catchment area larger than the grid area as “large lentic systems”, and linked them to the main channel corridor; correspondingly, the remaining lentic systems with a smaller upstream area are classified as “small lentic systems”, and linked to the subnetwork corridor.

We explained a definition of large or small lentic systems in the revised main text when the words firstly appeared. The specific modifications are found in L104-110.

L104-110: “Derived from two global lentic system datasets (HydroLAKES and GRanD database) (24, 25), we categorize lakes and reservoirs into “small” or “large”, depending on their upstream catchment area and the connectivity to subnetwork flows or main river channels in this study. Those with an upstream catchment area greater than the area of a 0.5° grid cell are defined as large lentic systems, and linked to the main channel corridor; correspondingly, the remaining lentic systems with a smaller upstream area are defined as small lentic systems, and linked to the subnetwork corridor.”

Reference:

24. Messenger, M. L., Lehner, B., Grill, G., Nedeva, I. & Schmitt, O. Estimating the volume and age of water stored in global lakes using a geo-statistical approach. *Nature Communications* 7, 13603 (2016).
25. Lehner, B., et al. High-resolution mapping of the world's reservoirs and dams for sustainable river-flow management. *Frontiers in Ecology and the Environment* 9, 494-502 (2011).

5. Comments: *L 157-174: It is unclear if the effect of changes of the surface area of lakes in response of climate change are accounted in the simulations (Zhao et al. 2022; Zhou et al. 2021).*

Response: In our study, we used the lake surface area data derived from the HydroLAKES dataset to drive the model. Due to the limitations of remote sensing products, this dataset can be considered as the most robust one to represent hydrological characteristics of the global lake systems despite it is static. Nevertheless, the changes of lake surface area due to climate change may result in non-negligible effects on lacustrine N₂O emissions in the real world. We further addressed this research gap in the Discussion section. The specific statement is found in L410-415.

L410-415: “Although we included additional N₂O emissions from newly constructed reservoirs over time, we did not consider the impact of lake area changes on N₂O emissions due to the limited availability of dynamic lake surface data. Considering that a recent study demonstrated the increasing trend of global lake area in recent decades (60), it is likely that our study gives a conservative estimate for N₂O-LR.”

Reference:

60. Pi, X., et al. Mapping global lake dynamics reveals the emerging roles of small lakes. *Nature Communications* 13, 5777 (2022).

6. Comments: *L 265: The two cited references are for a single system. It is unreasonable to make a generalization on the response of N₂O emissions from lentic systems to warming based on two sites.*

Response: We have revised the statement in the main text as:

L305-309: “Additionally, several local observational studies indicate that temperature increases are likely to stimulate the nitrifying and denitrifying microbial activity in water systems and increased

N₂O emissions (39, 40). It is worth noting that the increase in N₂O-LR caused by changes in terrestrial N loads may outweigh the effect of temperature on the control of microbial metabolism (39, 41)”

Reference:

39. Beaulieu, J. J., Shuster, W. D. & Rebolz, J. A. Nitrous oxide emissions from a large, impounded river: The Ohio River. *Environ. Sci. Technol.* 44, 7527-7533 (2010).
40. Xiao, Q., et al. Coregulation of nitrous oxide emissions by nitrogen and temperature in China's third largest freshwater lake (Lake Taihu). *Limnology and Oceanography* 64, 1070-1086 (2018).
41. Royer, T. V., David, M. B. & Gentry, L. E. Timing of riverine export of nitrate and phosphorus from agricultural watersheds in Illinois: Implications for reducing nutrient loading to the Mississippi River. *Environ. Sci. Technol.* 40, 4126-4131 (2006).

7. Comments: *L 275-296: Borges et al. (2022) suggested that N₂O in lakes could change as function of depth that modulates the coupling between surface waters and sediments. Since there is a general relation between surface area and depth (larger systems are in general deeper) (Wetzel 2001) this could also play a role in explaining differences between small and large systems.*

Response: Thank you for the valuable suggestion. The study by Borges et al. (2022) described the high denitrification rates in sediments of shallow African lakes, which supports the higher N₂O emission rates from the small lakes simulated in our study. We have added the standpoint in revised main text. The specific modifications are found in L337-341.

L337-341: “The shallow African lakes, with considerable organic matter deposition on the sediment, sustain high benthic denitrification rates, as suggested by Borges, et al. (17). Since there is a general relation between lake surface area and depth (52), that standpoint also supports the higher N₂O emission rates from the small lakes in our study.”

Reference:

17. Borges, A. V., et al. Greenhouse gas emissions from African lakes are no longer a blind spot. *Sci. Adv.* 8, eabi8716 (2022).
52. Wetzel, R. G. *Limnology: lake and river ecosystems* (gulf professional publishing) (2001).

8. Comments: *L 316-317: There have been several past papers reporting that the IPCC EF are too high. It would be useful to include these in the discussion here.*

Response: We have revised the statement in the main text as:

L379-382: “...we find that our results yield lower estimates than those reported in most of previous studies (7, 11, 14, 28) and align with the lower boundary of the range estimated by Maavara, et al. (8) (SI Appendix, Table S5).”

Reference:

7. Beaulieu, J. J., et al. Nitrous oxide emission from denitrification in stream and river networks. *Proc. Natl. Acad. Sci. U. S. A.* 108, 214-219 (2011).
8. Maavara, T., et al. Nitrous oxide emissions from inland waters: Are IPCC estimates too high? *Glob. Change Biol.* 25, 473-488 (2019).
11. Zheng, Y., et al. Global methane and nitrous oxide emissions from inland waters and estuaries. *Glob. Change Biol.* 28, 4713–4725 (2022).

14. IPCC 2019 Refinement to the 2006 IPCC Guidelines for National Greenhouse Gas Inventories Volume 4 Agriculture, Forestry and Other Land Use. (2019).
28. Tian, L., Cai, Y. & Akiyama, H. A review of indirect N₂O emission factors from agricultural nitrogen leaching and runoff to update of the default IPCC values. *Environ. Pollut.* 245, 300-306 (2019).

9. Comments: *L 340: Unfortunately the authors did not provide a Table with fluxes from observational studies. I have the impression the data from African lakes from the recent paper of Borges et al. (2022) are missing from this comparison.*

Response: We apologize for the oversight in not including detailed information regarding the observations used for model validation. We have added the three tables (SI Appendix, Table S7, Table S8, and Table S9) regarding the detailed information including location, sampled year, collected method, land use types, N₂O fluxes, aquatic nitrate concentration, N loads, and data sources of observation in Supplementary information. We also appreciate your valuable suggestion. We have incorporated the observed data from Borges et al., 2022 for validation (SI Appendix, Fig. S9), and the relevant detailed information can be found in Table S7 and Table S8.

10. Comments: *L 342: the r² alone is not sufficient for validation. You can have a very good correlation (high r²) and a poor validation due to a very large off-set (Y-intercept > or < 0) or over-/under-estimate (slope > or < 1) or both. So please make much more rigorous statistical evaluation of the validation.*

Response: Thank you for the valuable suggestion. We incorporated Nash-Sutcliffe efficiency coefficient (referred to as NSE hereafter) (Nash and Sutcliffe, 1970) as an additional metric, to enhance the reliability assessment of the simulations. NSE is considered to be a robust and objective quality measure and is widely used for assessing the goodness of fit of a variety of models, especially for hydrologic models (Clarke, 2008; McCuen et al., 2006; Lin et al., 2017). In the updated validation (SI Appendix, Fig. S9), the simulated N₂O emissions and aquatic nitrate concentrations for lakes, reservoirs, and rivers were comparable to observed values with R² values higher than 0.6 and NSEs higher than 0.5 in most cases. Additionally, we also plotted a 1:1 line on the validated figures to help identifying over- or underestimations.

Reference:

- Clarke R T. A critique of present procedures used to compare performance of rainfall-runoff models. *Journal of hydrology*, 2008, 352(3-4): 379-387.
- Lin F, Chen X, Yao H. Evaluating the use of Nash-Sutcliffe efficiency coefficient in goodness-of-fit measures for daily runoff simulation with SWAT. *Journal of Hydrologic Engineering*, 2017, 22(11): 05017023.
- McCuen R H, Knight Z, Cutter A G. Evaluation of the Nash–Sutcliffe efficiency index[J]. *Journal of hydrologic engineering*, 2006, 11(6): 597-602.
- Nash J E, Sutcliffe J V. River flow forecasting through conceptual models part I—A discussion of principles. *Journal of hydrology*, 1970, 10(3): 282-290.

SI Fig. S9 The location map of observations (a) and the comparisons of simulated inland water N₂O emissions (b, c, d) and aquatic nitrate concentrations (e, f, g) with observations. The sources of observed data used to validate inland water N₂O emissions and aquatic nitrate concentrations are provided in Tables S7 and S8.

11. Comments: L 346: *There's some room for debate. Borges et al. (2022) shows on the contrary that smaller and shallower African lakes have lower N₂O concentration than larger and deeper ones.*

Response: Thank you for the suggestion. The study by Borges et al. (2022) provides us with very useful information. We added the discussion for the different findings between our study and that by Borges et al. (2022). The specific modifications are found in L337-345.

L337-345: “The shallow African lakes, with considerable organic matter deposition on the sediment, sustain high benthic denitrification rates, as suggested by Borges, et al. (17). Since there is a general relation between lake surface area and depth (52), that standpoint also supports the higher N₂O emission rates from the small lakes in our study. However, if the inorganic nitrogen concentration

cannot support high denitrification rates on sediment, the N₂O produced in the water column will be absorbed by sediment to fuel benthic denitrification (17). This explains the contrasting findings in Borges, et al. (17) which showed the N₂O undersaturation in the shallow African lakes.”

Reference:

17. Borges, A. V., et al. Greenhouse gas emissions from African lakes are no longer a blind spot. *Sci. Adv.* 8, eabi8716 (2022).

52. Wetzel, R. G. *Limnology: lake and river ecosystems* (gulf professional publishing) (2001).

12. Comments: *L 348: do you mean seasonal or long-term changes of surface area?*

Please note that the HydroLAKES data-base in some cases over-estimates the lake surface area because some of data are quite old (HydroLAKES builds on the GLWD) and some lakes have shrunk with global warming. This is the case for arid and semi-arid climates, for instance Lake Chad, as discussed by Borges et al. (2022).

Response: The input data of lake surface area in our study is static. We concur with your point that global changes have indeed exerted a discernible influence on the hydrological characteristics of lakes. Consequently, the employment of data confined to a single period may introduce a degree of error into our estimations—a factor we have comprehensively addressed as a source of research uncertainty within the Discussion section.

A recent study conducted the estimate of the changes in global natural lake areas over the past few decades (Pi et al., 2022). The findings revealed that the area of natural lakes increased both within and outside the permafrost regions during the 1980s-2000s; however, during the 2000s-2010s, the area of natural lakes within the permafrost regions increased, while the area of natural lakes outside the permafrost regions decreased. Overall, the study demonstrated a net increase in the area of natural lakes across the six continents over the entire period. Incorporating this expansion of lake surface area could potentially result in a more substantial magnitude of actual N₂O emissions from lakes, indicating that our estimation of the increase in lake N₂O emissions may be underestimated.

Reference:

Pi X, Luo Q, Feng L, et al. Mapping global lake dynamics reveals the emerging roles of small lakes[J]. *nature communications*, 2022, 13(1): 5777.

We added the following discussion to the revised main text:

L413-415: “Considering that a recent study demonstrated the increasing trend of global lake area in recent decades (60), it is likely that our study gives a conservative estimate for N₂O-LR.”

Reference:

60. Pi, X., et al. Mapping global lake dynamics reveals the emerging roles of small lakes. *Nature Communications* 13, 5777 (2022).

13. Comments: *REFERENCES*

Borges AV, L Deirmendjian, S Bouillon, W Okello, T Lambert, FAE Roland, VF Razanamahandry, NRG Voarintsoa, F Darchambeau, IA Kimirei, J-P Descy, GH Allen, C Morana (2022) Greenhouse gas emissions from African lakes are no longer a blind spot, Science Advances, 8, eabi8716, 1-17, <https://doi.org/10.1126/sciadv.abi8716>

Chiriboga G & AV Borges (2023) Andean headwater and piedmont streams are hot spots of carbon dioxide and methane emissions in the Amazon basin, Communications Earth & Environment, Commun Earth Environ 4, 76 <https://doi.org/10.1038/s43247-023-00745-1>

Lauerwald R., P. Regnier, V. Figueiredo, A. Enrich-Prast, D. Bastviken, B. Lehner, T. Maavara, P. Raymond, Natural lakes are a minor global source of N₂O to the atmosphere. Global Biogeochem. Cycles, 33, 1564-1581 (2019).

Wetzel, R.G. (2001) Limnology Lake and Reservoir Ecosystems. Academic Press, San Diego.

Zhao G et al. (2022) Nat Commun 13, 3686, <https://doi.org/10.1038/s41467-022-31125-6>

Zhou W et al. (2021) Commun Earth Environ 2, 255, <https://doi.org/10.1038/s43247-021-00327-z>

Response: We sincerely appreciate your effort in providing us with these valuable references. Your contribution plays a crucial role in enhancing the quality of our manuscript. We have diligently included appropriate citations in the relevant statement.

Reviewer #2 (Remarks to the Author):

This study presents global predictions of N₂O emissions from lakes and reservoirs using a process based, coupled terrestrial-aquatic model. They find that while large lakes dominate lentic N₂O emissions, smaller lakes and small reservoirs have bigger per unit area emission. The study predicts smaller lentic N₂O emissions than most previous studies, but more than double the preindustrial emission rate.

A major strength of this study is the coupling of terrestrial and aquatic models that allow the exploration of how the major global change factors contribute to the increased N₂O emissions. The portioning of effects is very interesting. Not surprisingly, the major contributor is excess agricultural nitrogen. An interesting finding is that in recent decades, CO₂ fertilization seems to have led to reduced N₂O emissions, contributing to the slower increases in recent decades. This is based on the terrestrial model, which has its dynamics presumably described in earlier papers. It would be worth pointing out in this manuscript as part of discussion that, while aquatic N₂O emissions are predicted to be leveling off, no leveling off N₂O increases in the global atmosphere have been noted, instead they appear to be accelerating.

Response: We appreciate your positive comment on our work and the valuable suggestions that have significantly helped us to improve the manuscript. We also sincerely appreciate your suggestion regarding the different changing trends between N₂O emissions from global lentic systems and global atmospheric N₂O concentrations. However, we would like to clarify that changes in the atmospheric N₂O mixing ratio are not only related to changes in aquatic and land N₂O emissions, but are also influenced by atmospheric chemical reactions, which are way too much out of the scope of this study. Thus, we narrowed the scope of the discussion and added a statement about the different trends of aquatic and soil N₂O emission trends:

L326-328: “It would be worth pointing out that while N₂O-LR is predicted to be leveling off, no leveling off of increases in global soil N₂O emissions have been noted, instead they appear to be accelerating (1).”

Reference:

1. Tian, H., et al. A comprehensive quantification of global nitrous oxide sources and sinks. *Nature* 586, 248-256 (2020).

This manuscript emphasizes lentic N₂O emission, bringing in lotic N₂O emissions only as discussion points and as part of the emission factor calculation. This is surprising and I think a major weakness for several reasons. First, the results highlight the inland water emission factor, which includes streams and rivers, but their response to the global change factors studied in scenarios is not considered at all. The paper seems to be disconnected in this sense. Second, rivers actually dominate inland water N₂O emissions. Lentic waters are only about 10% of total inland water emissions. This is a point that I think is important to highlight. Third, river and lake processes interact as the authors discuss. However, the authors emphasize how denitrification by lakes limits N₂O emissions downstream, but not the fact that denitrification by streams and rivers limits N₂O emissions by lakes and reservoirs. This would be a compelling paper if these results were to be compared together.

Response: We greatly appreciate your constructive suggestions. In the revised main text, we provide in-depth information regarding the long-term changes and environmental impacts of global riverine N₂O emissions. The integrated N₂O emissions from inland waters as well as its portion caused by agricultural N additions is now also demonstrated in the revised main text. It is worth noting that the updated estimates of riverine N₂O emissions are lower compared to our previous report (Yao et al., 2020). As you pointed out, this discrepancy is attributed to the newly incorporated module for lentic systems, which intercepted a portion of nitrogen during transport. Meanwhile, rivers themselves intercept some N, thus decreasing the N load nitrified and denitrified in lakes and reservoirs. By conducting a comprehensive analysis that closely reflects the reality of the land to water continuum, we aim to achieve a more integrated understanding of N₂O emissions from inland waters. Our results emphasizing the dominant influence of agricultural N additions on riverine N₂O emissions, as well as highlights the significance of our work in providing country-specific agriculture-induced emission factors for inland water N₂O emissions.

Global inland water N₂O emissions are lower than most other studies, including those that incorporate the role of streams, rivers, lakes, and reservoirs, and that account for sub grid river networks (based on Beaulieu et al. 2011 which used the global aquatic N model in Wollheim et al. 2008). Little is discussed as to why the current model results in lower N₂O emissions. Is it because N loading from the landscape is less (Beaulieu model is based on annual N export from land to water in Green et al. 2004), the denitrification uptake velocity is less, the proportion of incomplete denitrification or nitrification is less, or some other reason. No parameter values are given (or calculated from the more complex model used here), but for the paper to be useful, they should be. Otherwise the results are just assertions.

Response: Our estimate for inland water N₂O emissions is only half of the estimate by Beaulieu et al. (2011). We infer that there are several reasons why that study reported a higher estimate. Firstly, their study estimated that 0.75% of anthropogenic dissolved inorganic nitrogen (DIN) inputs to river networks were converted to N₂O globally. This emissions factor is too high compared to the reported range in many recent studies (0.02%-0.26% according to studies by Hama-Aziz et al. (2016), Tian et al. (2019), Qin et al. (2019), and Hu et al. (2021)). Additionally, Beaulieu, et al. (2011) assumed that all N₂O produced in water is emitted, a point that has been argued due to its potential to overestimate aquatic N₂O emissions especially when considering water residence time. Furthermore, according to the emission factor of 0.75% and the total inland water N₂O emissions of 0.68 Tg N yr⁻¹, the magnitude of DIN input to river network would reach 90 Tg N yr⁻¹ in their study, which is clearly beyond the range of estimates from existing studies (18.8-50.3 Tg N yr⁻¹ for DIN loads according to Kroeze et al. (2005) and Hu et al. (2016)). Therefore, the study by Beaulieu et al. (2011) likely overestimated N₂O emissions from inland waters.

Reference:

- Hama-Aziz Z Q, Hiscock K M, Cooper R J. Indirect nitrous oxide emission factors for agricultural field drains and headwater streams[J]. Environmental Science & Technology, 2017, 51(1): 301-307.
- Tian L, Cai Y, Akiyama H. A review of indirect N₂O emission factors from agricultural nitrogen leaching and runoff to update of the default IPCC values[J]. Environmental pollution, 2019, 245: 300-306.
- Qin X, Li Y, Goldberg S, et al. Assessment of indirect N₂O emission factors from agricultural river networks based on long-term study at high temporal resolution[J]. Environmental Science & Technology, 2019, 53(18): 10781-10791.

Hu M, Li B, Wu K, et al. Modeling riverine N₂O sources, fates, and emission factors in a typical river network of eastern China[J]. Environmental Science & Technology, 2021, 55(19): 13356-13365.

Kroeze C, Dumont E, Seitzinger S P. New estimates of global emissions of N₂O from rivers and estuaries[J]. Environmental Sciences, 2005, 2(2-3): 159-165.

Hu M, Chen D, Dahlgren R A. Modeling nitrous oxide emission from rivers: a global assessment[J]. Global change biology, 2016, 22(11): 3566-3582.

The special modification regarding the comparison of estimate could be found in L288-295 of the revised main text. In the revised Supplementary Information, we have added the table (SI Appendix, Table S6) regarding the key parameters in the DLEM-TAC aquatic N₂O module used for the simulation on lentic systems.

L288-295: “Our estimated global inland water N₂O emission is only half of that estimated by Beaulieu et al. (7). Their study may have led to an overestimation (23), as their estimation was based on higher emission factors and dissolved inorganic N loads compared to those indicated by other studies (27-32). Furthermore, Beaulieu, et al. (7) assumed that all N₂O produced in water was emitted, a point that has been argued due to its potential to overestimate aquatic N₂O emissions especially when considering the effect of water residence time (8).”

Reference:

7. Beaulieu, J. J., et al. Nitrous oxide emission from denitrification in stream and river networks. Proc. Natl. Acad. Sci. U. S. A. 108, 214-219 (2011).

8. Maavara, T., et al. Nitrous oxide emissions from inland waters: Are IPCC estimates too high? Glob. Change Biol. 25, 473-488 (2019).

23. Yao, Y., et al. Increased global nitrous oxide emissions from streams and rivers in the Anthropocene. Nature Climate Change 10, 138-142 (2020).

27. Hama-Aziz, Z. Q., Hiscock, K. M. & Cooper, R. J. Indirect nitrous oxide emission factors for agricultural field drains and headwater streams. Environ. Sci. Technol. 51, 301-307 (2017).

28. Tian, L., Cai, Y. & Akiyama, H. A review of indirect N₂O emission factors from agricultural nitrogen leaching and runoff to update of the default IPCC values. Environ. Pollut. 245, 300-306 (2019).

29. Qin, X., et al. Assessment of indirect N₂O emission factors from agricultural river networks based on long-term study at high temporal resolution. Environ. Sci. Technol. 53, 10781-10791 (2019).

30. Hu, M., et al. Modeling riverine N₂O sources, fates, and emission factors in a typical river network of eastern China. Environ. Sci. Technol. 55, 13356-13365 (2021).

31. Kroeze, C., Dumont, E. & Seitzinger, S. P New estimates of global emissions of N₂O from rivers and estuaries. Environmental Sciences 2, 159-165 (2005).

32. Hu, M., Chen, D. & Dahlgren, R. A. Modeling nitrous oxide emission from rivers: a global assessment. Glob. Change Biol. 22, 3566-3582 (2016).

I did not understand the numbers that went into the global emission factor calculation. Please explain clearly how the 137.02 Gg N yr⁻¹ was derived.

Response: By the factorial experiments, we distinguished the effects of agricultural N addition on aquatic N₂O emissions. We found that agricultural N addition resulted in a substantial increase of 137 Gg N yr⁻¹ in inland water N₂O emissions in the 2010s. This value was used to calculate the contemporary agriculture-induced N₂O emission factors for inland waters. To address the specific

effects of agricultural N additions on N₂O emissions from global inland waters, we have included a table (SI Appendix, Table S2) in the revised Supplementary Information.

Another weakness is that the representation of the processes that lead to N₂O emissions and key parameter values are not provided. It is mentioned (L91) that significant improvements have been incorporated, but these are not identified. In the current manuscript, only the equation for mass balance of N₂O is provided, and variable, Y_{water}, is the only mention of N₂O production. There is nothing about what controls N₂O production, which is essential for a paper like this. I had to go back to the Yao et al. 2020 paper, where more detail is given in the supplemental, but here there were errors in description of the equations (supplemental eqns 5 -9) that I was not sure what was going on. No values for uptake velocity were given here, so the reader could not evaluate why the N₂O emissions were lower than other global studies. Maybe I could go back even further to earlier papers, but I should not have to as this is essential for understanding the results of this study.

It was also not clear how the lakes at the 0.5 degree grid scale were integrated into the river network from the description. Are all small lakes within a grid cell combined into one and “placed” at the mouth of the grid cell, intercepting N that has been transported through the sub grid cell river network where denitrification and N₂O emissions can also occur? It is likely that N enters streams at the sub grid scale first, before reaching any lake at the minimum lake sizes considered. See Wollheim et al. 2008 for a very similar approach (Wollheim, W.M., C.J. Vorosmarty, A.F. Bouwman, P.A. Green, J. Harrison, E. Linder, B.J. Peterson, S. Seitzinger, and J.P.M. Syvitski. 2008b. Global N removal by freshwater aquatic systems: a spatially distributed, within-basin approach. Global Biogeochemical Cycles. GB2026, doi:10.1029/2007GB002963.

Response: To address the ambiguity surrounding the representative processes and key parameters regarding aquatic N₂O emission in our model, we have added the detailed description in the revised Supplementary Information (SI Appendix, Text S1-S4, Table S6). Furthermore, we have elucidated the depiction of lentic systems and their interconnectedness with river systems within the model.

I appreciated the effort at validation of N₂O emissions, based on 60 lake locations, as well as the incorporation of uncertainty in the loading, which I agree is an essential issue that must be considered. But I would like to know where these observational data come from. There were no units in the figures showing these comparisons with observations. Given that this is a global model, and that nitrate concentrations likely drive N₂O emissions, in order to understand the usefulness of the model I would also like to see how well the model is tested against nitrate concentration data in streams, rivers, or lakes. Since the complete N cycle is represented, it would also be useful to compare the % of incomplete denitrification predicted to measurements, which based on LINX2 studies (Beaulieu et al 2011) demonstrated a fairly low average of 0.9% of denitrification is as N₂O. Are predictions here similar?

Response: We appreciate your positive comments. In the revised Supplementary Information, we provided detailed information on the observed data, including the location, sampled time, collected method, land use types, N₂O emission, N loads, and data sources (SI Appendix, Table S7 and Table S9). Furthermore, we added the validation of aquatic nitrate concentrations in streams, rivers, lakes,

and reservoirs. The relevant information on the observed nitrate concentrations is provided as well (SI Appendix, Table S8). In the modified validation figure, we additionally showed the spatial distribution of the observed data as well as the units (SI Appendix, Fig. S9).

In our model, we set the ratio of N₂O produced from incomplete denitrification as 1% (as shown in SI Appendix, Table S6), which is in the range provided in the previous study (McCrackin et al., 2010) and is closed to the ratio in the LINX2 study.

Reference:

McCrackin M L, Elser J J. Atmospheric nitrogen deposition influences denitrification and nitrous oxide production in lakes[J]. Ecology, 2010, 91(2): 528-539.

I also wondered about the mechanisms behind a few of the results. Why does Africa have such large increase in N₂O emissions, while ag N increase is minimal. Change is attributed to climate change in Africa, but I think it is getting drier, so why would that lead to more N₂O emissions? Table S1 reports the residence times of lakes and reservoirs. Why would small lakes have a longer residence time than large lakes (I don't think this is expected), and do reservoirs really have 10's of years of residence time?

Response: Based on the results of factorial experiments, we have demonstrated that climate change and land use change are the dominant factors driving the changes of lentic N₂O emissions from Africa (see SI Appendix, Fig. S3). Due to rapid population growth in sub-Saharan Africa, particularly in humid regions, there has been a significant increase in demand for agricultural land and forest products, resulting in the deforestation of large areas of natural forests (Brandt et al., 2017). The conversion of these natural forests which provide soil protection and bonding functions, to other artificial land use types has accelerated nutrients loss from soil through erosion, leading to soil degradation (Vågen et al., 2005). Furthermore, more frequent occurrence of extreme precipitation was observed in Africa (Thackeray et al., 2022), which would exacerbate the impact of land use management shifts on soil nutrients loss. The discussion regarding the changes of the lentic N₂O emissions from Africa has been included in the revised main text:

L309-319: “Land use conversions play a significant role in altering soil N cycling and promoting terrestrial N losses (42, 43), thereby contributing to increased N₂O-LR. This phenomenon appears to be particularly pronounced in Africa. With rapid population growth and increased demand for agricultural land and wood products, large areas of natural forests in sub-Saharan Africa have been deforested or converted to agricultural land (44). The conversion of natural forests which serve the function of protecting and bonding soils, to other artificial land-use types caused severe nutrients loss from soil (45), leading to a significant increase in N₂O-LR. In addition, the increasing frequency of extreme precipitation events observed in Africa (46) has accelerated the soil erosion by water on these human-disturbed regions.”

Reference:

42. Fu, B., et al. Effects of land use on soil erosion and nitrogen loss in the hilly area of the Loess Plateau, China. Land Degradation Development 15, 87-96 (2004).

43. Wuaden, C. R., Nicoloso, R. S., Barros, E. C. & Grave, R. A. Early adoption of no-till mitigates soil organic carbon and nitrogen losses due to land use change. Soil Tillage Research 204, 104728 (2020).

44. Brandt, Martin, et al. Human population growth offsets climate-driven increase in woody vegetation in sub-Saharan Africa. Nature Ecology & Evolution, 0081 (2017).

45. Vågen, T-G, Lal, Rattan & Singh, BR Soil carbon sequestration in sub-Saharan Africa: a review. *Land Degradation & Development* 16, 53-71 (2005).
46. Thackeray, Chad W, Hall, Alex, Norris, Jesse & Chen, Di Constraining the increased frequency of global precipitation extremes under warming. *Nature Climate Change* 12, 441-448 (2022).

The HydroLAKES dataset, where the lake attribute data used to drive our model was derived from, have demonstrated that water residence time increases with lake surface area (Messenger et al., 2016). In order to meet the model input requirements in our study, we process this dataset into the data with 0.5-degree spatial resolution, which means that the lake attribute on a specific grid is the comprehensive attribute data of all lakes covered by this grid. We realized that the water residence time in SI Appendix, Table S2 caused misunderstanding to readers, so we revised the table in the new version.

Reference:

- Thackeray C W, Hall A, Norris J, et al. Constraining the increased frequency of global precipitation extremes under warming[J]. *Nature Climate Change*, 2022, 12(5): 441-448.
- Brandt M, Rasmussen K, Peñuelas J, et al. Human population growth offsets climate-driven increase in woody vegetation in sub-Saharan Africa[J]. *Nature ecology & evolution*, 2017, 1(4): 0081.
- Vågen T G, Lal R, Singh B R. Soil carbon sequestration in sub-Saharan Africa: a review[J]. *Land degradation & development*, 2005, 16(1): 53-71.
- Messenger M L, Lehner B, Grill G, et al. Estimating the volume and age of water stored in global lakes using a geo-statistical approach[J]. *Nature communications*, 2016, 7(1): 13603.

Specific

Response: Please check the detailed revision and point-by-point responses that follow.

1. Comments: *L97. Define small and large lakes.*

Response: In accordance with your suggestion along with Comment#4 from Reviewer #1, we explained a definition of large or small lentic systems in main text when these terms firstly appeared. The specific modifications are found in L104-110.

L104-110: “Derived from two global lentic system datasets (HydroLAKES and GRanD database) (24, 25), we categorize lakes and reservoirs into “small” or “large”, depending on their upstream catchment area and the connectivity to subnetwork flows or main river channels in this study. Those with an upstream catchment area greater than the area of a 0.5° grid cell are defined as large lentic systems, and linked to the main channel corridor; correspondingly, the remaining lentic systems with a smaller upstream area are defined as small lentic systems, and linked to the subnetwork corridor.”

Reference:

24. Messenger, M. L., Lehner, B., Grill, G., Nedeva, I. & Schmitt, O. Estimating the volume and age of water stored in global lakes using a geo-statistical approach. *Nature Communications* 7, 13603 (2016).
25. Lehner, B., et al. High-resolution mapping of the world's reservoirs and dams for sustainable river-flow management. *Frontiers in Ecology and the Environment* 9, 494-502 (2011).

2. Comments: *L106. I would suggest one fewer decimal place when reporting fluxes. I'm sure estimates are not that precise, and would make easier to read.*

Response: We have revised the reported fluxes by rounding them to one decimal place in the text, figures, and tables. Given the small magnitude of regional N₂O emissions from small lentic systems, we retain two decimal places for the increasing rates of N₂O emissions from small lentic systems in Europe, North America, and East Asia in L164-167 of the main text.

L164-167: “The increase in rates of N₂O emission from small lentic systems in Europe (0.03 Gg N yr⁻¹), North America (0.02 Gg N yr⁻¹), and East Asia (0.04 Gg N yr⁻¹) were much higher than those in other regions.”

3. Comments: *L162. It was not clear what this statement means.*

Response: We apologize for the unclear statement. Although we used static water surface area data as input for the model, the initiation of reservoir simulations on the grid cells is determined by the year of reservoir construction. Specifically, the simulation of the reservoir process takes place only after a reservoir has been constructed on a specific grid cell. Consequently, the number of reservoirs included in the simulation increases as reservoirs are constructed during the simulation period. The same strategy for reservoir is also included in the simulation of scenario experiments, suggesting that the effects of environmental factors on N₂O emissions from lentic systems encompass not only those caused by environmental changes in the catchment, but also a small fraction of the effects associated with reservoir construction and thus the increase of reservoir quantity and of overall water residence time in the inland water network. (Given that N₂O emissions from reservoirs constituted 12% of the total N₂O emissions from lentic systems in the 2010s, we assume that the impact of reservoir construction represents only a minor portion of the overall impact).

We have rewritten the description of the impacts of reservoir construction for greater clarity. The specific modifications are found in L171-177.

L171-177: “It should be noted that the initiation of reservoir simulations on the grid cells is determined by the year of reservoir construction. As a result, when conducting factorial experiments, the effect of reservoir construction will undoubtedly be included in the influence of environmental changes on N₂O-LR. Since N₂O emissions from reservoirs constituted 12% of N₂O-LR in the 2010s, we assume that the impact of reservoir construction represents only a minor portion of the overall impact.”

4. Comments: *L192. Why do smaller systems have a greater response in the model?*

Response: We infer that the size of the small lentic systems and its geographic location contribute to the greater response of its N₂O emissions to environmental changes. Small lentic systems tend to have lower buffering capacity to environmental change due to the smaller surface areas and volumes (Choi, 1998). The relative importance of the changes in terrestrial N inputs increases relative to lake area and volume, hence having a greater response on N₂O emissions from small lentic systems.

Reference:

Choi, J. S., 1998. Lake ecosystem responses to rapid climate change. *Environmental Monitoring and Assessment* 49: 281–290.

We have added the explanation in revised main text. The specific modifications are found in L354-357.

L354-357: “Compared to large lentic systems, small lentic systems are characterized by higher importance of terrestrial N inputs relative to surface area and volume. Therefore, changes in terrestrial N inputs caused by global change showed a stronger impact on N₂O emissions from small lentic systems.”

5. Comments: *L195. Confusing section here.*

Response: We apologize for the confusing statement. Through conducting factorial experiments, we investigated the impact of different environmental changes on N₂O emissions from lentic systems. Our analysis showed that from the 1850s to the 2010s, the proportion of responses attributed to small lentic systems surpassed the proportion of total area they occupy (36%) in most of cases (SI Appendix, Fig. S4). This suggested that small lentic systems have greater response to environmental changes.

Furthermore, our findings reveal that the environmental variables that result in a strong response of small lentic systems were not consistent across the three simulation periods. Specifically, during the 1850s-1940s, the strong responses of N₂O emissions from small lentic systems were attributable to climate change (63%), agricultural N addition (90%), atmospheric N deposition (72%), and increased atmospheric CO₂ concentration (-54%), and outweighed the responses of large lentic systems, constituting more than half of the total responses (SI Appendix, Fig. S4(a)). During the 1940s-1980s, while the influence of climate change on N₂O emissions from small lentic systems diminished (decreasing from 63% to 36%), the responses of small lentic systems to agricultural N additions (68%), atmospheric N deposition (54%), and increased atmospheric CO₂ concentration (-64%) still amounted to over half of the total responses. In the period of the 1980s- 2010s, N₂O emissions from small lentic systems have greater responses to natural disturbances such as climate change (56%) and increased atmospheric CO₂ concentration (-63%).

We have rewritten the initial text for greater clarity. The specific modifications are found in L207-218.

L207-218: “Specifically, during the 1850s-1940s, the strong responses of N₂O emissions from small lentic systems were attributable to climate change (63%), agricultural N addition (90%), atmospheric N deposition (72%), and increased atmospheric CO₂ concentration (-54%), and outweighed the responses of large lentic systems, constituting more than half of the total responses. During the 1940s-1980s, while the influence of climate change on N₂O emissions from small lentic systems diminished (decreasing from 63% to 36%), the responses of small lentic systems to agricultural N additions (68%), atmospheric N deposition (54%), and increased atmospheric CO₂ concentration (-64%) still amounted to over half of the total responses. In the period of the 1980s- 2010s, N₂O emissions from small lentic systems have greater responses to natural disturbances such as climate change (56%) and increased atmospheric CO₂ concentration (-63%) (SI Appendix, Fig. S4).”

SI Fig. S4 The contributions of environmental factors to the changes of N₂O emissions from small lentic systems (a) and large lentic systems (b) in different time periods. The black bars show mean decadal emissions induced by five environmental factors. The percent changes between the time periods indicate the net change of N₂O emissions induced by five environmental factors during the corresponding periods. The colored bars represent the contribution of each environmental factor to the net change of N₂O emissions from small or large lentic systems for the corresponding periods. The colored percentages represent the relative contribution of response of N₂O emission from small or large lentic systems to the total response of N₂O emissions under the influence of each environmental factor. Ag-N addition includes nitrogen fertilizer and manure application.

6. Comments: *L203. What about the role of streams and rivers upstream of lakes? Should be mentioned.*

Response: Thanks for your suggestion. The significance of riverine hydrological processes in relation to aquatic CO₂ emission is underscored by the study conducted by Liu et al. (2022). According to the results from that study, riverine discharge plays a pivotal role in modulating the impact of terrestrial carbon sources on aquatic CO₂ emissions. Consequently, the riverine CO₂ emissions are subject to variation, commensurate with changes in riverine discharge. The similar effect exists for the terrestrial-aquatic N cycle; whereby the transport of N in inland waters is regulated by discharge. In this regard, upstream rivers act as conduits, transporting N to lakes, where it participates in nitrification and denitrification processes.

Reference:

Liu S, Kuhn C, Amatulli G, et al. The importance of hydrology in routing terrestrial carbon to the atmosphere via global streams and rivers. *Proceedings of the National Academy of Sciences*, 2022, 119(11): e2106322119.

We have added a description of the role of upstream streams and rivers on nutrient N transportation to the revised main text. The specific modifications are found in L221-223.

L221-223: “For instance, the nutrient N is transported to lakes and reservoirs through upstream rivers; as a potent reactor of N species, lakes or reservoirs have a significant impact on nutrient N exported further to the downstream rivers.”

7. Comments: *L210. This is only half of the contemporary estimate from the global model of lakes, rivers, reservoirs reported in Beualeigh et al. 2011. Can you include some discussion of why?*

Response: We guess you mean the study by Beaulieu et al. (2011), which estimated that the global river network, including lakes and reservoirs, emitted 680 Gg N yr⁻¹ of N₂O to the atmosphere. In that paper, the fractions of DIN input emitted as N₂O to atmosphere by denitrification were fixed at 0.25% based on observations of 72 headwater streams in North America. Subsequently, their study assumed that nitrification converts twice as much anthropogenic DIN to N₂O as denitrification (e.g., 0.5%). Finally, they estimated that 0.75% of anthropogenic DIN inputs to river networks were converted to N₂O globally. This emissions factor is too high compared to the reported range in many recent studies (0.02%-0.26% according to studies by Hama-Aziz et al. (2016), Tian et al. (2019), Qin et al. (2019), and Hu et al. (2021)). Furthermore, according to the emission factor of 0.75% and the total inland water N₂O emissions of 0.68 Tg N yr⁻¹, the magnitude of DIN input to river network would reach 90 Tg N yr⁻¹ in their study, which is clearly beyond the range of estimates from existing studies (18.8-50.3 Tg N yr⁻¹ for DIN loads according to Kroeze et al. (2005) and Hu et al. (2016)).

In addition, Beaulieu et al. (2011) assume that all N₂O produced in river networks is emitted, a point that has been argued due to its potential to overestimate N₂O emissions from rivers especially when considering water residence time (Maavara et al., 2019). Therefore, the study by Beaulieu et al. (2011) likely overestimated N₂O emissions from inland waters.

Reference:

Beaulieu J J, Tank J L, Hamilton S K, et al. Nitrous oxide emission from denitrification in stream and river networks. *Proceedings of the National Academy of Sciences*, 2011, 108(1): 214-219.

Hama-Aziz Z Q, Hiscock K M, Cooper R J. Indirect nitrous oxide emission factors for agricultural field drains and headwater streams[J]. *Environmental Science & Technology*, 2017, 51(1): 301-307.

Tian L, Cai Y, Akiyama H. A review of indirect N₂O emission factors from agricultural nitrogen leaching and runoff to update of the default IPCC values[J]. *Environmental pollution*, 2019, 245: 300-306.

Qin X, Li Y, Goldberg S, et al. Assessment of indirect N₂O emission factors from agricultural river networks based on long-term study at high temporal resolution[J]. *Environmental Science & Technology*, 2019, 53(18): 10781-10791.

Hu M, Li B, Wu K, et al. Modeling riverine N₂O sources, fates, and emission factors in a typical river network of eastern China[J]. *Environmental Science & Technology*, 2021, 55(19): 13356-13365.

Kroeze C, Dumont E, Seitzinger S P. New estimates of global emissions of N₂O from rivers and estuaries[J]. *Environmental Sciences*, 2005, 2(2-3): 159-165.

Hu M, Chen D, Dahlgren R A. Modeling nitrous oxide emission from rivers: a global assessment[J]. *Global change biology*, 2016, 22(11): 3566-3582.

Maavara T, Lauerwald R, Laruelle G G, et al. Nitrous oxide emissions from inland waters: Are IPCC estimates too high?. *Global change biology*, 2019, 25(2): 473-488.

We have included the discussion in the revised main text. The specific modifications are found in L288-295.

L288-295: “Our estimated global inland water N₂O emission is only half of that estimated by Beaulieu, et al. (7). Their study may have led to an overestimation (23), as their estimation was based on higher emission factors and dissolved inorganic N loads compared to those indicated by other studies (27-32). Furthermore, Beaulieu, et al. (7) assumed that all N₂O produced in water was emitted, a point that has been argued due to its potential to overestimate aquatic N₂O emissions especially when considering the effect of water residence time (8).”

Reference:

7. Beaulieu, J. J., et al. Nitrous oxide emission from denitrification in stream and river networks. *Proc. Natl. Acad. Sci. U. S. A.* 108, 214-219 (2011).

8. Maavara, T., et al. Nitrous oxide emissions from inland waters: Are IPCC estimates too high? *Glob. Change Biol.* 25, 473-488 (2019).

23. Yao, Y., et al. Increased global nitrous oxide emissions from streams and rivers in the Anthropocene. *Nature Climate Change* 10, 138-142 (2020).

27. Hama-Aziz, Z. Q., Hiscock, K. M. & Cooper, R. J. Indirect nitrous oxide emission factors for agricultural field drains and headwater streams. *Environ. Sci. Technol.* 51, 301-307 (2017).

28. Tian, L., Cai, Y. & Akiyama, H. A review of indirect N₂O emission factors from agricultural nitrogen leaching and runoff to update of the default IPCC values. *Environ. Pollut.* 245, 300-306 (2019).

29. Qin, X., et al. Assessment of indirect N₂O emission factors from agricultural river networks based on long-term study at high temporal resolution. *Environ. Sci. Technol.* 53, 10781-10791 (2019).

30. Hu, M., et al. Modeling riverine N₂O sources, fates, and emission factors in a typical river network of eastern China. *Environ. Sci. Technol.* 55, 13356-13365 (2021).

31. Kroeze, C., Dumont, E. & Seitzinger, S. P New estimates of global emissions of N₂O from rivers and estuaries. *Environmental Sciences* 2, 159-165 (2005).

32. Hu, M., Chen, D. & Dahlgren, R. A. Modeling nitrous oxide emission from rivers: a global assessment. *Glob. Change Biol.* 22, 3566-3582 (2016).

8. Comments: *L247. Lake emissions should also be lower when incorporated into river network because of stream and river reductions. i.e. it works in both directions.*

Response: Thanks for your suggestion. We have included the discussion in the revised main text. The specific modifications are found in L283-284.

L283-284: “Simultaneously, modeling of in-river transformation and losses of N will also restrict N transfer to lentic systems.”

9. Comments: *L277. Are small lakes hotspots not just because more effective denitrification (uptake velocity / hydraulic load), but also because they are upstream from large lakes, and therefore have*

greater inputs? It is not clear why residence times should be greater in small lakes, as it is likely that they will also have less watershed area. Please support this is case from literature.

Response: Yes, we agree with your standpoint. Previous studies have confirmed the greater N removal in small lakes (Cheng and Basu, 2017; Harrison et al., 2009). Moreover, owing to their advantageous geographic position (are upstream from large lakes), small lakes could intercept a sizable portion of terrestrial N loads prior to reaching larger downstream lakes, thus preventing this intercepted N from participating in the nitrification and denitrification processes in larger downstream lakes.

Reference:

Cheng F Y, Basu N B. Biogeochemical hotspots: Role of small water bodies in landscape nutrient processing[J]. *Water Resources Research*, 2017, 53(6): 5038-5056.

Harrison J A, Maranger R J, Alexander R B, et al. The regional and global significance of nitrogen removal in lakes and reservoirs[J]. *Biogeochemistry*, 2009, 93: 143-157.

We revised the relevant statement in the main text as follow:

L332-337: “This can be attributed not only to the higher effectiveness of small lentic systems in N removal processes (51), but also to their advantageous geographic position, enabling them to intercept a sizable portion of terrestrial N loads prior to reaching downstream large lentic systems. Consequently, this interception prevents the captured N from contributing to the nitrification and denitrification occurring in the downstream large lentic systems.”

Reference:

51. Harrison, J. A., et al. The regional and global significance of nitrogen removal in lakes and reservoirs. *Biogeochemistry* 93, 143-157 (2009).

10. Comments: *L283. Longer residence times lead to more denitrification, but not more complete conversion of nitrate to N₂, at least based on this reference. This statement requires more support.*

Response: Thank you for the suggestion. Longer water residence time raise the opportunity for denitrification and N₂O emissions in large reservoirs. However, the N retention by upstream reservoirs reduces the substrate available for nitrification and denitrification within downstream large reservoirs. As reported in previous studies, upstream reservoirs can reduce more than 50% of NO₃-N inputs to the downstream reservoirs. This explains the lower emission rate of large reservoirs in our study. We have revised the statement in the main text. The specific modifications are found in L347-354.

L347-354: “Although previous work has highlighted the contribution of longer water residence times and an anoxic bottom water column on promoting denitrification within large reservoirs (54), it is crucial to recognize the significant role played by upstream small reservoirs in limiting denitrification in downstream large reservoirs. Their effective retention of inorganic N substantially reduced N concentrations in the outflow (reductions can even exceed 50%) (55-58), thereby restricting the nitrification and denitrification within downstream large reservoirs.”

Reference:

54. Liang, X., et al. Control of the hydraulic load on nitrous oxide emissions from cascade reservoirs. *Environmental Science & Technology* 53, 11745-11754 (2019).

55. David, M. B., Wall, L. G., Royer, T. V. & Tank, J. L. Denitrification and the nitrogen budget of a reservoir in an agricultural landscape. *Ecological Applications* 16, 2177-2190 (2006).
56. Shaughnessy, A. R., Sloan, J. J., Corcoran, M. J. & Hasenmueller, E. A. Sediments in agricultural reservoirs act as sinks and sources for nutrients over various timescales. *Water Resources Research* 55, 5985-6000 (2019).
57. Stenback, G. A., Crumpton, W. G. & Schilling, K. E. Nitrate loss in Saylorville Lake reservoir in Iowa. *Journal of Hydrology* 513, 1-6 (2014).
58. Schilling, K. E., Anderson, E., Streeter, M. T. & Theiling, C. Long-term nitrate-nitrogen reductions in a large flood control reservoir. *Journal of Hydrology* 620, 129533 (2023).

11. Comments: L292. *Is this really an important dynamic? If so, provide some numbers.*

Response: In the revised main text, we ascribed the greater response of N₂O emissions in small lentic systems to their smaller volume and surface area, making them more susceptible to alterations in terrestrial N inputs driven by environmental changes. For instance, equivalent alterations of terrestrial N input would lead to more pronounced fluctuations in nitrate concentration within small lentic systems as opposed to larger ones, thereby influencing the denitrification.

L354-357: “Compared to large lentic systems, small lentic systems are characterized by higher relative importance of terrestrial N inputs relative to surface area and volume. Therefore, changes in terrestrial N inputs caused by global change showed a stronger impact on N₂O emissions from small lentic systems.”

12. Comments: L297. *Please be clear that this includes streams and rivers, in addition to lentic waters. Since this is highlighted, stream and river results should be added to results section.*

Response: Great suggestion. We have added the estimate for the N₂O emissions from global streams and rivers in the subsection “**Estimate of global inland water N₂O emissions: integrating updated riverine estimation**”.

L219-241: “**Estimate of global inland water N₂O emissions: integrating updated riverine estimation.** The transport and transformation of N in inland waters have significant cascading effects. For instance, the nutrient N is transported to lakes and reservoirs through upstream rivers; as a potent reactor of N species, lakes or reservoirs have a significant impact on nutrient N exported further to the downstream rivers. Therefore, in this study, we present an updated estimation of N₂O emissions from global streams and rivers, encompassing the processes of lentic systems that were excluded from our previous estimates (SI Appendix, Fig. S5). In the pre-industrial period (the 1850s), global streams and rivers emitted 83.8 ± 22.8 Gg N yr⁻¹ of N₂O into atmosphere. Since the pre-industrial era, there has been a significant growth in global riverine N₂O emissions, particularly from the 1960s to the 2010s, exhibiting a linear growth rate of 26.2 Gg N per decade (SI Appendix, Fig. S5). The remarkable increase of global riverine N₂O emissions can be attributed primarily to the application of agricultural N fertilizer and N manure (SI Appendix, Fig. S6). In the recent decade, global riverine N₂O emissions reached 254.9 ± 46.2 Gg N yr⁻¹, marking a twofold increase from the estimated levels in the 1850s (SI Appendix, Fig. S5).

Combining the updated amount of riverine N₂O emissions, the estimated N₂O emission from global inland waters in the 2010s was 319.6±58.2 Gg N yr⁻¹ (Fig. 4). In the 2010s, riverine N₂O emissions constitute 80% of the total emissions from inland waters worldwide, with N₂O-LR being accountable for the remaining 20%. N₂O emissions from global inland waters have substantially increased by 207.2 Gg N yr⁻¹ since the 1850s, of which the increased N₂O-LR induced by anthropogenic perturbation contributes to 17% of the total increase of N₂O emissions from inland waters.”

SI Fig. S5 Dynamic N₂O emissions from global streams and rivers during the 1850s-2010s.

13. Comments: L315-317. *This discounting is unclear.*

Response: The conventional definition of N₂O emission factor for inland water in the previous studies was defined as the proportion of N₂O emissions to the aquatic nitrate content or leached agricultural N addition. In contrast, our study defined the emission factor as the proportion of agricultural-induced N₂O emissions from inland waters to the agricultural N addition to the land. To facilitate a seamless comparison of our emission factor with the previously computed ones, we incorporated a 24% adjustment for terrestrial N loss through leaching and runoff which derived from the IPCC’s report (IPCC, 2019), to discount the emission factors in previous studies.

Reference:

IPCC. 2019 Refinement to the 2006 IPCC Guidelines for National Greenhouse Gas Inventories Volume 4 Agriculture, Forestry and Other Land Use, 2019.

14. Comments: L331-346. *Consider putting model validation in a brief study design paragraph before the results, to increase trust in the model results.*

Response: We appreciate your suggestion. We added the model validation in the last paragraph of Introduction section. The specific modifications are found in L98-101.

L98-101: “We compared the simulated inland water N₂O fluxes and aquatic nitrate concentration with the observations around the globe to showcase the good performance of our model, with R² values exceeding 0.6 and Nash-Sutcliffe efficiency coefficient (NSE) exceeding 0.5.”

15. Comments: L371. *Did the sub grid cell river network account for serial transformations from 1st through 4th order? The integration of lakes is unclear – do small lakes intercept all sub grid river flow?. The global inland water N model of Wollheim et al. 2008 used a similar approach, including a statistical N removal model to account for small river dynamics.*

Response: In the DLEM-TAC, we didn’t consider the serial transformations within small streams. We refer to MOSART river transport model in our aquatic module, which synthesizes the rivers of 1st-5th order within the 0.5° grid cell as a virtual river channel to simulate the transport processes. For more details, please refers to Li et al., 2013.

Reference:

Li H, Wigmosta M S, Wu H, et al. A physically based runoff routing model for land surface and earth system models. *Journal of Hydrometeorology*, 2013, 14(3): 808-828.

Sub grid lakes do not intercept all flow of sub grid rivers in DLEM-TAC. We allocated the water and nutrient flows of sub grid rivers based on the upstream area of sub grid lakes and reservoirs. Only the water and nutrient flows corresponding to the upstream area of the sub grid lakes and reservoirs could flow into the sub grid lakes and reservoirs.

Regarding the study by Wollheim et al. (2008), as you pointed out, they used a statistical approach to achieve a relatively simple simulation of N removal in small river dynamics. The MOSART river transport model provides a physical mechanism of the water transporting. However, the MOSART model does not implement a complex description of the serial transformations from 1st through 5th order of rivers.

According to your suggestion, we added description of the water and nutrient allocations to sub grid lakes and reservoirs in the revised main text. The specific modifications are found in L447-450.

L447-450: “The incoming water and nutrient flows of sub grid lakes and reservoirs linked to subnetworks depend on their upstream area obtained from the high-resolution dataset (24), which determines the fraction of flows from hillslope and subsurface that are intercepted.”

Reference:

24. Messenger, M. L., Lehner, B., Grill, G., Nedeva, I. & Schmitt, O. Estimating the volume and age of water stored in global lakes using a geo-statistical approach. *Nature Communications* 7, 13603 (2016).

16. Comments: L384. *If lakes and reservoir residence times are predefined, does that mean it does not change with changes in hydrology (i.e. climate change)?*

Response: The utilization of predefined outflow rates based on water residence times is a widely adopted approach in global-scale modeling studies (Coe et al., 2000). In addition, our study employed

a high-precision lake dataset that incorporates spatially varying water residence times, offering a distinct advantage over some previous studies that relied on globally fixed values. Even so, we encourage the future development of high temporal precision lake hydrologic dataset to help reduce uncertainty in the modeling studies, as stated in Discussion section.

Reference:

Coe M T. Modeling terrestrial hydrological systems at the continental scale: Testing the accuracy of an atmospheric GCM. *Journal of Climate*, 2000, 13(4): 686-704.

17. Comments: *L396. More in how Y_{water} is controlled is really needed.*

Response: We sincerely apologize for the unclear statement. We have added the details for Y_{water} in Supplementary Information. The specific modifications are found in SI Appendix, Text S2.

Supplementary Text S2: The simulation of N_2O production within inland waters Y_{water}

Dissolved N_2O production in water column was calculated from both nitrification and denitrification:

$$Y_{water} = R_{nitrif} \times k_{nitrif} + R_{denitrif} \times k_{denitrif} \quad (12)$$

where k_{nitrif} and $k_{denitrif}$ are the nitrogen (N) removal rate ($g\ N\ d^{-1}$) through nitrification and denitrification, respectively. R_{nitrif} and $R_{denitrif}$ are the associated ratio of N_2O production through nitrification and denitrification, respectively (4). The nitrification or denitrification rate (k) can be estimated as:

$$k = \exp\left(\frac{-v}{\Delta d}\right) \quad (13)$$

where v is the settling velocity (d^{-1}) of NO_3^- or ammonia (NH_4^+) through nitrification or denitrification, respectively, and Δd is the hydraulic load (m) for water flow into the downstream grid cell. v can be simulated by a first-order kinetics equation (5):

$$v = v_{ref} (Q_{10})^{\frac{T-T_s}{10}} \quad (14)$$

where v_{ref} is the settling velocity of NO_3^- or NH_4^+ at the reference temperature of 20 °C, Q_{10} is the change fraction of NO_3^- or NH_4^+ reaction rates at a temperature change of 10 °C and assigned 2.0 here, T is the water temperature (°C), and T_s is the reference temperature (20 °C). Δd can be expressed as:

$$\Delta d = \frac{Q}{A_s} \quad (15)$$

where Q is discharge ($m\ s^{-1}$), and A_s is the surface area of the waterbody.

Reference:

4. Beaulieu, J. J., et al. Nitrous oxide emission from denitrification in stream and river networks. *Proc. Natl. Acad. Sci. U. S. A.* 108, 214-219 (2011).
5. Yang, Q., et al. Increased nitrogen export from eastern North America to the Atlantic Ocean due to climatic and anthropogenic changes during 1901–2008. *Journal of Geophysical Research: Biogeosciences* 120, 1046-1068 (2015).

18. Comments: *L422. Unclear what “dam operation closed” means. Is it that there are no reservoirs at all, or just input = output, or there is active management for hydroelectricity based on some rules?*

Response: The objective of the simulation with dam operation closed is to establish the base flow in natural river networks in the absence of reservoirs. We provided additional explanations in the main text:

L498-499: “Then we conducted the natural flow simulation with the dam model temporarily deactivated (no dams)...”

19. Comments: *L449. Does this mean that 76% of added N is harvested, stored, or denitrified on land? Isn't this part of the terrestrial model used here?*

Response: The IPCC's report reviewed a range of literature and demonstrated an average loss ratio of 0.24 for agricultural N addition through leaching and runoff. We refer to the loss ratio in the text to discount the emission factors in previous studies as the proportion of aquatic N₂O emissions relative to land N additions and facilitate comparison with our emission factor at the same level. We agree with your standpoint that the remaining 76% of N addition is allocated to crop harvesting, storage, and N-related gas emissions from soil according to the assumptions in the IPCC's report.

In our terrestrial module, we didn't use a constant ratio to fix the proportion of N fertilizer lost by leaching and runoff. Instead, we considered N uptake by vegetation and impacts of soil conditions on N transforms within the grid cell. As a result, different input conditions on each grid cell can lead to changes in the proportion of N addition to be lost.

20. Comments: *L479. The data availability is all raw input data from other sources, none of the data produced by this study. I would expect key model predictions would be shared, as well as the model validation data that was compiled.*

Response: Thank you for the comment. We will share key predictions as well as validation data along with the article publication.

21. Comments: *Fig. 4. Color reference are confusing, as red represents both lake and changing emissions.*

Response: We have adjusted the colors of Fig. 4 for better readability.

Fig. 4 Global inland water N_2O emissions for the 2010s. The colored arrows represent N_2O emissions as follows: green, emissions from streams and rivers; blue, emissions from small and large lakes; pink, emissions from small and large reservoirs. The colored numbers represent N_2O fluxes as follows: bold black numbers, the emissions in the 2010s; bold and italic red numbers, the increased emissions during the 1850s-2010s.; numbers with green, blue, and pink background colors represent total N_2O emissions from rivers, lakes, and reservoirs, respectively. The unit for all numbers is $Gg\ N\ yr^{-1}$.

22. Comments: Fig 5b. Change secondary Y axis to “proportion manure”.

Response: We have revised the secondary Y axis for Fig. 5(b).

Fig. 5 Dynamics of mean global EFs for inland water N_2O emissions (a) and the proportion of manure in agricultural N additions (b).

23. Comments: Table S2. Sort based on Country name, or EFag.

Response: We have ranked SI Appendix, Table S3 in descending order according to the values of the emission factors.

24. Comments: *Table S4. I'm not sure this table is worth including, given the very different assumption, with mist not relevant to the IPCC report.*

Response: Thanks for your suggestion. We have revised the table to encompass previous emission factors based on the same assumptions. This modification should improve the reader's comprehension of the emission factor presented in our study. Accordingly, we have revised the related descriptions in L377-382 of the main text.

L377-382: “After discounting emission factors collected from previous studies based on 24% of the proportion leached from agricultural N additions (14), we find that our results yield lower estimates than those reported in most of previous studies (7, 11, 14, 28) and align with the lower boundary of the range estimated by Maavara, et al. (8) (SI Appendix, Table S5).”

Reference:

7. Beaulieu, J. J., et al. Nitrous oxide emission from denitrification in stream and river networks. *Proc. Natl. Acad. Sci. U. S. A.* 108, 214-219 (2011).
8. Maavara, T., et al. Nitrous oxide emissions from inland waters: Are IPCC estimates too high? *Glob. Change Biol.* 25, 473-488 (2019).
11. Zheng, Y., et al. Global methane and nitrous oxide emissions from inland waters and estuaries. *Glob. Change Biol.* 28, 4713–4725 (2022).
14. IPCC 2019 Refinement to the 2006 IPCC Guidelines for National Greenhouse Gas Inventories Volume 4 Agriculture, Forestry and Other Land Use. (2019).
28. Tian, L., Cai, Y. & Akiyama, H. A review of indirect N₂O emission factors from agricultural nitrogen leaching and runoff to update of the default IPCC values. *Environ. Pollut.* 245, 300-306 (2019).

SI Table S5 The emission factors of N₂O emissions from inland waters published in the literatures.

Original				
Regions	Types	Relative to	EFs %	References
Global	Inland waters	nitrate concentration	0.25	(17)
Global	Inland waters	N loads	0.75	(4)
Global	Inland waters	nitrate concentration	0.26	(18)
Global	Inland waters [†]	nitrate concentration	0.83	(12)
Global	River and Reservoir	N loads	0.18-0.45	(14)
Discounted based on a 24% loss of agricultural N addition through runoff and leaching, as recommended by the IPCC report (17)				
Regions	Types		EFs %	References

Global	Inland waters	0.062	(17)
Global	Inland waters	0.225 [§]	(4)
Global	Inland waters	0.062	(18)
Global	Inland waters	0.199	(12)
Global	River and Reservoir	0.043-0.108	(14)

Note:

‡ The types of water systems include streams, rivers, lakes, and reservoirs.

§ the value is discounted by the loss fraction of 30% as assumed in that study.

25. Comments: *Figure S2. Make legend larger to read*

Response: We have revised the Fig. S2 in the Supplementary Information.

SI Fig. S2 The long-term changes of regional N₂O-*LR* (NA: North America; SA: South America; EU: Europe; RUS: Russia; AF: Africa; SAS: South Asia; EAS: East Asia; SEAS: Southeast Asia; W/CAS: West/Central Asia; OCE: Oceania).

26. Comments: *Table S4. CO₃ should have negative values, right?*

Response: Yes, the effect of atmospheric CO₂ concentration should have negative values in the period of the 1980s to the 2010s. We apologize for the inaccurate representation in the Fig. S4 of

Supplementary Information. We have corrected the value for the effect of atmospheric CO₂ concentration during the 1980s-2010s.

SI Fig. S4 The contributions of environmental factors to the changes of N₂O emissions from small lentic systems (a) and large lentic systems (b) in different time periods. The black bars show mean decadal emissions induced by five environmental factors. The percent changes between the time periods indicate the net change of N₂O emissions induced by five environmental factors during the corresponding periods. The colored bars represent the contribution of each environmental factor to the net change of N₂O emissions from small or large lentic systems for the corresponding periods. The colored percentages represent the relative contribution of response of N₂O emission from small or large lentic systems to the total response of N₂O emissions under the influence of each environmental factor. Ag-N addition includes nitrogen fertilizer and manure application.

27. Comments: *Figure S5 and S8. Units needed, more informative caption (where are data from, how collected, ecosystem, biome, time). References to where they are from, and include in data that this study makes available.*

Response: Thanks for your valuable suggestion. We have updated the validated results for N₂O emissions from three water types, aquatic nitrate concentrations, and terrestrial N loads in SI Appendix, Fig.S9 and Fig.S10. In addition, the detailed information of the observation, including the location, sampled time, collected method, land use types, N₂O emission, N loads, and data sources, has provide in the SI Appendix, Table S7-S9.

SI Fig. S9 The location map of observations (a) and the comparisons of simulated inland water N₂O emissions (b, c, d) and aquatic nitrate concentrations (e, f, g) with observations. The sources of observed data used to validate inland water N₂O emissions and aquatic nitrate concentrations are provided in Tables S7 and S8.

SI Fig. S10 Comparisons of simulated terrestrial N loads with observations. The sources of observed data are provided in Tables S9.

Reviewer #3 (Remarks to the Author):

I have published quite a lot concerning GHG emissions from lakes so the literature is well-known to me. N₂O is a minor issue in lakes as it probably constitutes less than 3% of lakes' atmospheric impact so that does diminish the interest of this manuscript somewhat. I have a major concern about this manuscript that would need to be cleared up before publication anywhere, in my view. This paper is basically a lengthy modeling effort base on the premise that N limits N₂O emissions from lakes. The authors cite a couple of manuscripts as support of this idea (Maavara et al. 2018 [not 2019 as listed] and Lauerwald et al 2019) but both of these are modeling exercises and neither of them demonstrates the critical role of N in driving N₂O emissions from lakes. Instead, it is much more likely (see DelSontro et al 2018) that N₂O emissions are driven by P (or primary production) and lake size. Most limnologists agree that N is almost always in excess in inland waters so the amount of N as substrate for N₂O emission should rarely be an issue. A credible study would be based on empirical analyses of the factors driving N₂O emissions in lakes. In its present state, this paper appears to be a thought study based on the query, "what would be happening if N limited N₂O emissions".

Some smaller suggestions are listed below

Response: Thank you for the note. As Taylor Maavara and Ronny Lauerwald, co-authors of the current manuscript, clarified, the Maavara et al. paper in *Global Change Biology* was accepted and first appeared online as an uncorrected preprint proof in late 2018, but the copy-edited, typeset version that appeared in an issue of the journal was published in February 2019. Hence, 2019 is the correct and official citation of record that is indexed by the journal itself, Scopus, ORCID, etc.

The reviewer also seems to have misunderstood the modeling approach used in Maavara et al., 2019, and Lauerwald et al., 2019, as the models presented in these papers do account for N vs. P limitation in controlling primary productivity, as well as situations in which light limitation governs productivity instead of a nutrient limitation, which is calculated based on organic carbon content in the water column. The models in both papers are calibrated using a wealth of empirical and mechanistic literature that shows that N₂O emissions are controlled by the amount of available nitrogen. We additionally note that the reviewer is possibly confusing the role that nutrient limitation plays in controlling primary productivity (which is well understood, and we agree that freshwater lakes are usually P limited), with N₂O production, which occurs through nitrification and denitrification, which are different processes.

We are familiar with the findings in DelSontro et al., 2018, where they empirically conclude that N₂O emissions are somewhat related to lake size and chl-a. Single paper is insufficient to provides enough evidence that N₂O emissions are more likely controlled by P availability than N availability, and the study by Maavara et al. (2019) and Lauerwald et al. (2019) do not apply this approach in the model for the following reasons:

- (1) Despite what the reviewer has said best predicted N₂O emissions, DelSontro et al 2018 did not show that P was the best predictor of N₂O emissions. They fit empirical relationships between N₂O emissions and total N, total P, lake surface area, and chl-a. In fact, they found a better relationship with TN than TP as a predictor of N₂O emissions, which is also what is shown in most existing literature, including the references we cite throughout our manuscript

to justify our decision to have N₂O be related to the amount of N in the system. Furthermore, DelSontro et al's most robust empirical relationship to predict N₂O emissions was with chl-a and SA, and used data from 268 lakes had an R² of 0.09, meaning this relationship only predict 9% of the variance in their data. We find it difficult to statistically justify using such a weak correlation to conclusively draw such a confident global conclusion as the reviewer does in their comment above regarding the role of any parameter in N₂O emissions, especially as their conclusions about the relationship between photosynthesis and N₂O emissions have not been shown in other studies.

- (2) The model we present is mechanistic, i.e. it represents processes that control N₂O production and emissions based on processes in the form of biogeochemical reactions that can be represented with kinetics. N₂O is produced during denitrification and nitrification, i.e. reactions that convert nitrate to N₂ gas, and ammonium to nitrate, respectively, with N₂O produced as a byproduct or reaction intermediate. Primary production, meanwhile, refers to processes that use nitrate and ammonium to generate organic molecules, and does not produce N₂O as intermediates or byproducts. Hence, attempting to represent N₂O emissions mechanistically using mechanisms that do not produce N₂O is not possible. This represents a good example of the limitation of empirical models in general: while there may be a (weak) relationship between N₂O emissions and primary production, the empirical data itself does not help us better understand the specific drivers at play, and there is no robust way to test with the dataset available whether the correlation is spurious.

We have addressed the rest of the reviewer's specific comments in our detailed revision of the manuscript and point-by-point responses that follow.

1. Comments: *L55-56. There are lots of studies of N₂O emissions in lakes and reservoirs. This statement is false or misleading.*

Response: We have corrected the statement in the revised main text:

L52-53: "However, the global estimates are still weakly constrained, particularly for lentic systems such as lakes and reservoirs."

2. Comments: *L229-230. Ranges of estimated impacts are listed as N here, not as CO₂ equivalents as in most similar studies.*

Response: Thank you for the suggestion. We have incorporated the corresponding CO₂-equivalent values into the text.

L262-266: "In the past, observation-based studies have provided a rough referred range for N₂O-LR (160.0-380.0 Gg N yr⁻¹, 30.0-70.0 Gg N yr⁻¹, and 400.9 Gg N yr⁻¹ for lakes, reservoirs, and total lentic systems, which were equivalent to 173.3-326.0 Tg N yr⁻¹, 25.7-60.1 Tg N yr⁻¹, and 344.0 Tg N yr⁻¹ of CO₂ emissions, respectively, using a GWP of 273 over 100 years; see SI Appendix, Table S4)."

3. Comments: L232-234: “Observation-based” studies are suggested to be inaccurate compared to this model. Science normally depends on observations so the authors should make the case why their model might actually be more accurate. This seems to be somewhat non-scientific, to me.

Response: Observed studies are the basis for model construction and development. In this study, we also require observational data for model calibration and validation. However, most of observation-based studies were strongly dependent on observations with low data coverage and additionally geographically skewed, to obtain global estimates for lentic N₂O emissions by the average of measured areal N₂O fluxes and the total surface water area (e.g., Deemer et al. (2016) utilized the observations from 57 systems globally; Soued et al. (2016) utilized the observations from 157 systems globally, 121 of which were from small oligotrophic lakes in Iran). These studies provide a reference for the future estimate, in the other hand, will introduce errors due to scale effects by neglecting the influence of spatial heterogeneity of the basin conditions on lentic N₂O emissions. Such scale effects may be more pronounced in the case of sparse spatial distribution of measured data.

4. Comments: L371-372. Quite a lot is known about the global distribution of rivers and streams. For example, Downing et al. 2012. Why not use the best available estimates? In any case, I think the authors should explain why stream order (which is often criticized as having little biogeochemical relevance) would be relevant.

Response: There are actually no stream order in our model, but we refer to the stream order typically included in a 0.5° grid cell to help reader understand the concept of small streams at sub grid level. We rely on the two studies by Fekete et al. (2001) and Vörösmarty et al. (2000) to explain that small rivers within a 0.5° grid cell generally represent the 1st-5th stream orders. In our model, we represent all rivers within the 0.5° grid cell as a virtual river channel. For more details, please refer to our response to comment#16 by reviewer#2. We appreciate your suggestion regarding the study by Downing et al. (2012), but their analysis for rivers isn't applicable to our simulation.

Reference:

Fekete, B.M., Vörösmarty, C.J. and Lammers, R.B., 2001. Scaling gridded river networks for macroscale hydrology: Development, analysis, and control of error. *Water Resources Research*, 37(7), pp.1955-1967.

Vörösmarty, C.J., Fekete, B.M., Meybeck, M. and Lammers, R., 2000. A simulated topological network representing the global system of rivers at 30-minute spatial resolution (STN-30). *Global Biogeochemical Cycles*, 14, pp.599-621.

5. Comments: L 394. The authors should better explain where variables like F_a or Y_{water} could be determined and where these came from.

Response: We sincerely apologize for the unclear statement. We have added the details for F_a and Y_{water} in Supplementary Information. The specific modifications are found in SI Appendix Text S1 and Text S2.

Supplementary Text S1: The simulation of advective N₂O flux F_a

In the sub-grid level, the inflow and outflow rates of river channels, lakes and reservoirs in the model can be described by the following equations:

$$\frac{\partial Q_{channel,out}}{\partial x} + L \frac{\partial h}{\partial t} = Q_{channel,in} \quad (1)$$

$$Q_{sublake,in} = Q_{hill} \times (Area_{upstream,sublake} \div Area_{grid}) \quad (2)$$

$$Q_{subreservoir,in} = Q_{hill} \times (Area_{upstream,subreservoir} \div Area_{grid}) \quad (3)$$

$$Q_{subnetwork,in} = Q_{hill} - Q_{sublake,in} - Q_{subreservoir,in} + Q_{sublake,out} + Q_{subreservoir,out} \quad (4)$$

Where $Q_{channel,out}$ and $Q_{channel,in}$ are outflow rates and inflow rates of river channels (note that the label “channel” can be hillslope, subnetwork, and main channel, respectively); L the length of the river channels; h is the depth of runoff; note that the outflow rates of river channels shown in the equation (1) are solved by Kinematic Wave Method in our model (1); $Q_{subnetwork,in}$, $Q_{sublake,in}$, and $Q_{subreservoir,in}$ are inflow rates of subnetworks, small lakes, and small reservoirs ($m^3 d^{-1}$), respectively; $Q_{sublake,out}$ and $Q_{subreservoir,out}$ are outflow rates of small lakes and small reservoirs ($m^3 d^{-1}$), which are calculated from the water residence time of lakes and reservoirs obtained from HydroLAKES dataset; Q_{hill} are outflow rates of hillslope; $Area_{grid}$, $Area_{upstream,sublake}$, $Area_{upstream,subreservoir}$ are the areas of the grid cell, the upstream of small lakes and small reservoirs, respectively.

Accordingly, the advective N_2O fluxes through subnetworks ($F_{a,subnetwork}$), small lakes ($F_{a,sublake}$) and small reservoirs ($F_{a,subreservoir}$) are described as:

$$F_{a,sublake} = Q_{sublake,in} \times C_{hill} - Q_{sublake,out} \times C_{sublake,out} \quad (5)$$

$$F_{a,subreservoir} = Q_{subreservoir,in} \times C_{hill} - Q_{subreservoir,out} \times C_{subreservoir,out} \quad (6)$$

$$F_{a,subnetwork} = Q_{subnetwork,in} \times C_{hill} + Q_{sublake,out} \times C_{sublake,out} + Q_{subreservoir,out} \times C_{subreservoir,out} + Y_{g/h} - Q_{subnetwork,out} \times C_{subnetwork,out} \quad (7)$$

where $Q_{subnetwork,out}$ are inflow rates of subnetworks ($m^3 d^{-1}$); C_{hill} , $C_{subnetwork,out}$, $C_{sublake,out}$, and $C_{subreservoir,out}$ are concentrations ($g N m^{-3}$) of dissolved N_2O in the outflow of hillslope (C_{hill} equals to the equilibrium N_2O concentration), subnetworks, small lakes, and small reservoirs, respectively; We assumed that the dissolved N_2O yield ($Y_{g/h}$ in $g N d^{-1}$) in groundwater and hyporheic zones is linearly related to the land nitrate (NO_3^-) leaching rate:

$$Y_{g/h} = \sum r_{g/h} \times Loading_{NO_3^-} \times Area_{veg} \quad (8)$$

where $r_{g/h}$ is the ratio of N_2O production over the leached NO_3^- (unitless), $Loading_{NO_3^-}$ is the NO_3^- leaching rate from land ($g N m^{-2} d^{-1}$) calculated in DLEM-TAC, and $Area_{veg}$ is vegetation area (m^2) of different plant functional types.

The advective N_2O fluxes through the main channels ($F_{a,main}$), large lakes ($F_{a,largelake}$), and large reservoirs ($F_{a,largerreservoir}$) are described as:

$$F_{a,largelake} = Q_{up,out} C_{up,out} - Q_{largelake,out} C_{largelake,out} \quad (9)$$

$$F_{a,largereservoir} = Q_{largelake,out}C_{largelake,out} - Q_{largereservoir,out}C_{largereservoir,out} \quad (10)$$

$$F_{a,main} = Q_{largereservoir,out}C_{largereservoir,out} + Q_{sunetwork,out}C_{subnetwork,out} - Q_{main,out}C_{main,out} \quad (11)$$

where $Q_{up,out}$ and $Q_{main,out}$ are the outflow rates ($\text{m}^3 \text{d}^{-1}$) of upstream grid cells and the main channels in the current grid cell, which are solved by Kinematic Wave Method (equation (1)); $Q_{largelake,out}$ are outflow rates of large lakes ($\text{m}^3 \text{d}^{-1}$), which are calculated from the water residence time of lakes obtained from the HydroLAKES dataset; $Q_{largereservoir,out}$ are outflow rates of large reservoirs ($\text{m}^3 \text{d}^{-1}$), which are quantified through a dam operation module using natural flow, dam storage and dam type as model inputs (2, 3); $C_{up,out}$, $C_{main,out}$, $C_{largelake,out}$, $C_{largereservoir,out}$ are the associate N_2O concentrations (g N m^{-3}).

Supplementary Text S2: The simulation of N_2O production within inland waters Y_{water}

Dissolved N_2O production in water column was calculated from both nitrification and denitrification:

$$Y_{water} = R_{nitrif} \times k_{nitrif} + R_{denitrif} \times k_{denitrif} \quad (12)$$

where k_{nitrif} and $k_{denitrif}$ are the nitrogen (N) removal rate (g N d^{-1}) through nitrification and denitrification, respectively. R_{nitrif} and $R_{denitrif}$ are the associated ratio of N_2O production through nitrification and denitrification, respectively (4). The nitrification or denitrification rate (k) can be estimated as:

$$k = \exp\left(\frac{-v}{\Delta d}\right) \quad (13)$$

where v is the settling velocity (d^{-1}) of NO_3^- or ammonia (NH_4^+) through nitrification or denitrification, respectively, and Δd is the hydraulic load (m) for water flow into the downstream grid cell. v can be simulated by a first-order kinetics equation (5):

$$v = v_{ref}(Q_{10})^{\frac{T-T_s}{10}} \quad (14)$$

where v_{ref} is the settling velocity of NO_3^- or NH_4^+ at the reference temperature of 20°C , Q_{10} is the change fraction of NO_3^- or NH_4^+ reaction rates at a temperature change of 10°C and assigned 2.0 here, T is the water temperature ($^\circ\text{C}$), and T_s is the reference temperature (20°C). Δd can be expressed as:

$$\Delta d = \frac{Q}{A_s} \quad (15)$$

where Q is discharge ($\text{m}^3 \text{s}^{-1}$), and A_s is the surface area of the waterbody.

Reference:

1. Chow, V. T. (1964) A compendium of water-resources technology, Ven Te Chow. Handbook of Applied Hydrology), pp 8-61.

2. Biemans, H., et al. Impact of reservoirs on river discharge and irrigation water supply during the 20th century. *Water Resources Research* 47, (2011).
3. Lehner, B., et al. High-resolution mapping of the world's reservoirs and dams for sustainable river-flow management. *Frontiers in Ecology and the Environment* 9, 494-502 (2011).
4. Beaulieu, J. J., et al. Nitrous oxide emission from denitrification in stream and river networks. *Proc. Natl. Acad. Sci. U. S. A.* 108, 214-219 (2011).
5. Yang, Q., et al. Increased nitrogen export from eastern North America to the Atlantic Ocean due to climatic and anthropogenic changes during 1901–2008. *Journal of Geophysical Research: Biogeosciences* 120, 1046-1068 (2015).

6. Comments: *L 401. It should be clarified how one gets from modeled or simulated N transport to N₂O emissions. Especially given the paucity of global empirical data suggested in the Introduction.*

Response: Thank you for the comment. We have described the representation for the N transport to aquatic N₂O emissions within the DLEM-TAC in the previous statement. The specific description is found in L453-456.

L453-456: “The aquatic N module was developed based on the scale adaptive water transport scheme (23, 68), including lateral transport, decomposition of organic matter, particulate organic matter deposition, nitrification, and denitrification.”

Reference:

23. Yao, Y., et al. Increased global nitrous oxide emissions from streams and rivers in the Anthropocene. *Nature Climate Change* 10, 138-142 (2020).
68. Li, H., et al. Evaluating global streamflow simulations by a physically based routing model coupled with the community land model. *Journal of Hydrometeorology* 16, 948-971 (2015).

Reviewer #1 (Remarks to the Author):

The authors have made a wonderful job to address my comments from the previous round of review. I strongly suggest they incorporate a discussion on the differences in model set-up and differences of data output with the recent publication of Wang et al. (2023)

Wang J et al. (2023) Inland Waters Increasingly Produce and Emit Nitrous Oxide, *Environ. Sci. Technol.* 2023, 57, 36, 13506–13519, <https://doi.org/10.1021/acs.est.3c04230>

Reviewer #2 (Remarks to the Author):

Thank you for considering and incorporating many of my suggestions. I particularly appreciate the addition of the nitrate validation and table of parameter values. These are essential for interpreting these results compared to other studies. The paper is greatly improved as a result.

I think some additional thought and discussion of why these global estimates are on the low end of what other studies have reported is still needed however. It is fine for there to be differences, but these should be discussed clearly. Two reasons are invoked by the authors: N₂O emission factors (which emerge from denitrification uptake velocity and hydrologic conditions) and DIN loading rates. These are currently not well addressed. First, uptake velocity for lakes (Table S6) used here is much lower than in previous studies (e.g. Beaulieu uses LINX 2 empirical results, from headwater streams applied to all surface waters, while here it is assumed from another modeling study). I believe uptake velocity works out to be 10 fold lower here, which would explain much of the difference. This fact needs to be made explicit. The second is that loading rates are mentioned to be much lower here. However, the global input of DIN and DON to the river network is not provided to evaluate this. I recommend adding this to the discussion as well.

Finally, the authors should revisit their supplemental material to make sure they are representing their equations properly. For example, there are many errors in Supplemental Text 2. Alternatively, if this is what was used, that undermines all the results. Make sure units cancel out. I don't see how k is g/d based on eqn 13. Settling velocity needs units of m/d. so does hydraulic load. If a reader is supposed to understand why results here are lower than previous estimates, then currently it is not possible.

Revisions of Manuscript: NCOMMS-23-28044A

Title: Increased nitrous oxide emissions from global lakes and reservoirs since the pre-industrial era

Author(s): Ya Li, Hanqin Tian, Yuanzhi Yao, Hao Shi, Zihao Bian, Yu Shi, Siyuan Wang, Taylor Maavara, Ronny Lauerwald, Shufen Pan

We express our gratitude to the reviewers and editors for taking the time to read and review this manuscript as well as for their constructive comments and feedback. We have carefully revised the manuscript in line with the feedback from the reviewers, expanding on the discussions regarding the comparison of different estimates from our study and from others, along with the underlying reasons. Additionally, we have corrected the equations in the Supplemental Text 2 to ensure the units can be canceled out.

The revisions are outlined below with a point-by-point response to each reviewer's comments. Please note that reviewers' comments are in *italics* while our responses are not. All texts from the manuscript or Supplemental Information are colored blue, and citation numbers correspond to the reference section of this manuscript.

Point-to-Point Responses to Reviewers' Comments

Reviewer #1 (Remarks to the Author):

The authors have made a wonderful job to address my comments from the previous round of review. I strongly suggest they incorporate a discussion on the differences in model set-up and differences of data output with the recent publication of Wang et al. (2023).

Wang J et al. (2023) Inland Waters Increasingly Produce and Emit Nitrous Oxide, Environ. Sci. Technol. 2023, 57, 36, 13506–13519, <https://doi.org/10.1021/acs.est.3c04230>

Response: We appreciate your positive feedback and constructive suggestions for improving the quality of the manuscript. We have included the discussion into the main text to elucidate the difference in our estimate compared to that of Wang et al. (2023). The specific modifications are found in L301-310.

L301-310: “Another recent modelling study by Wang et al. (27) reporting higher inland water N₂O emissions of 0.4 Tg N yr⁻¹ in 1900 and 1.3 Tg N yr⁻¹ in 2010. However, in their study, the oversight of the seasonal emission fluctuations under the yearly modelling time step and the potential inhibited effect of elevated atmospheric CO₂ levels on terrestrial N availability and subsequent N loss may introduce significant uncertainty into their estimates. Moreover, for some N sources included in their study such as aquaculture and wastewater, the existing datasets still fall short in providing us with accurate quantification on the effect of these sources on global inland water N₂O emissions. Therefore, future development of modelling input data will help reduce uncertainties in the model estimates.”

Reference:

27. Wang, J., et al. Inland waters increasingly produce and emit nitrous oxide. Environ. Sci. Technol. **57**, 13506-13519 (2023).

Reviewer #2 (Remarks to the Author):

Thank you for considering and incorporating many of my suggestions. I particularly appreciate the addition of the nitrate validation and table of parameter values. These are essential for interpreting these results compared to other studies. The paper is greatly improved as a result.

Response: We would like to express our sincere gratitude for your positive feedback. Your valuable suggestions have not only enhanced the clarity of the manuscript but have also improved the importance of this work.

I think some additional thought and discussion of why these global estimates are on the low end of what other studies have reported is still needed however. It is fine for there to be differences, but these should be discussed clearly. Two reasons are invoked by the authors: N₂O emission factors (which emerge from denitrification uptake velocity and hydrologic conditions) and DIN loading rates. These are currently not well addressed. First, uptake velocity for lakes (Table S6) used here is much lower than in previous studies (e.g. Beaulieu uses LINX 2 empirical results, from headwater streams applied to all surface waters, while here it is assumed from another modeling study). I believe uptake velocity works out to be 10 fold lower here, which would explain much of the difference. This fact needs to be made explicit. The second is that loading rates are mentioned to be much lower here. However, the global input of DIN and DON to the river network is not provided to evaluate this. I recommend adding this to the discussion as well.

Response: Great suggestion. We have now included an explicit statement that the lower estimate in our study is primarily attributable to the lower denitrification uptake velocity and terrestrial nitrogen input to inland waters compared to those by Beaulieu et al. (2011). Furthermore, we have provided a detailed comparison of the parameter values. The specific modifications are found in L287-294.

Additionally, we have incorporated some additional discussion regarding the varying estimates from our study and those from two recent studies by Lauerwald et al. (2023) and Wang et al. (2023). The specific modifications are found in L299-310.

L287-294: “Our estimated global inland water N₂O emission is only half of that estimated by Beaulieu et al. (7), which may be attribute to the discrepancy on the uptake velocities and terrestrial N input to inland waters. For instance, we assumed the lower denitrification uptake velocity ranging from 3E-08 to 2E-06 m s⁻¹ and simulated lower DIN input of 48 Tg N yr⁻¹ (TN input of 89 Tg N yr⁻¹) in the 2010s, compared to their reported denitrification uptake velocity of 8E-08 to 1E-05 m s⁻¹ and DIN input of 90 Tg N yr⁻¹, respectively.”

L299-310: “A latest synthesis (28) homogenized global scale estimates to present N₂O emission of 204.9 (157.8-375.5) Gg N yr⁻¹ from global inland waters, which is close to our estimate. Another recent modelling study by Wang et al. (27) reporting higher inland water N₂O emissions of 0.4 Tg N yr⁻¹ in 1900 and 1.3 Tg N yr⁻¹ in 2010. However, in their study, the oversight of the seasonal emission fluctuations under the yearly modelling time step and the potential inhibited effect of elevated atmospheric CO₂ levels on terrestrial N availability and subsequent N loss may introduce significant uncertainty into their estimates. Moreover, for some N sources included in their study such as aquaculture and wastewater, the existing datasets still fall short in providing us with accurate

quantification on the effect of these sources on global inland water N₂O emissions. Therefore, future development of modelling input data will help reduce uncertainties in the model estimates.”

Reference:

7. Beaulieu, J. J., et al. Nitrous oxide emission from denitrification in stream and river networks. Proc. Natl. Acad. Sci. U. S. A. **108**, 214-219 (2011).
27. Wang, J., et al. Inland waters increasingly produce and emit nitrous oxide. Environ. Sci. Technol. **57**, 13506-13519 (2023).
28. Lauerwald, R., et al. Inland water greenhouse gas budgets for RECCAP2: 2. Regionalization and homogenization of estimates. Glob. Biogeochem. Cycle **37**, e2022GB007658 (2023).

Finally, the authors should revisit their supplemental material to make sure they are representing their equations properly. For example, there are many errors in Supplemental Text 2. Alternatively, if this is what was used, that undermines all the results. Make sure units cancel out. I don't see how k is g/d based on eqn 13. Settling velocity needs units of m/d. so does hydraulic load. If a reader is supposed to understand why results here are lower than previous estimates, then currently it is not possible.

Response: Thanks for your correction. We have carefully checked the equations in Supplemental Text 2, and have made the following revisions:

Supplementary Text S2: The simulation of N₂O production within inland waters Y_{water}

Dissolved N₂O production (g N d⁻¹) in water column was calculated from both nitrification and denitrification:

$$Y_{water} = R_{nitrif} \times k_{nitrif} \times C_{nhx} \times Q + R_{denitrif} \times k_{denitrif} \times C_{noy} \times Q \quad (12)$$

where k_{nitrif} and $k_{denitrif}$ are the nitrogen (N) removal efficiency (unitless) through nitrification and denitrification, respectively; R_{nitrif} and $R_{denitrif}$ are the associated ratio of N₂O production through nitrification and denitrification, respectively (4). C_{nhx} and C_{noy} are N contents of the water (g N m⁻³); Q is discharge (m³ d⁻¹); The nitrification or denitrification efficiency (k) can be estimated as:

$$k = \exp\left(\frac{-v}{\Delta d}\right) \quad (13)$$

where v is the settling velocity (m d⁻¹) of NO₃⁻ or ammonia (NH₄⁺) through nitrification or denitrification, respectively, and Δd is the hydraulic load (m d⁻¹) for water flow into the downstream grid cell. v can be simulated by a first-order kinetics equation (5):

$$v = v_{ref} (Q_{10})^{\frac{T-T_s}{10}} \quad (14)$$

where v_{ref} is the settling velocity of NO₃⁻ or NH₄⁺ at the reference temperature of 20 °C, Q_{10} is the change fraction of NO₃⁻ or NH₄⁺ reaction rates at a temperature change of 10 °C and assigned 2.0 here, T is the water temperature (°C), and T_s is the reference temperature (20 °C). Δd can be expressed as:

$$\Delta d = \frac{Q}{A_s} \quad (15)$$

where Q is discharge ($\text{m}^3 \text{d}^{-1}$), and A_s is the surface area of the waterbody.

Reviewer #2 (Remarks to the Author):

The authors have addressed all my comments. Nice work!